# From Evaluation to Design: Using Potential Energy Surface Smoothness Metrics to Guide ML Interatomic Potential Architectures

Ryan Liu [1] [2] [*]   Eric Qu [2]   Tobias Kreiman [2]   Samuel M Blau [3]   Aditi S. Krishnapriyan [2]

## Abstract

Machine Learning Interatomic Potentials (MLIPs) sometimes fail to reproduce the physical smoothness of the quantum potential energy surface (PES), leading to erroneous behavior in downstream simulations that standard energy and force regression evaluations can miss. Existing evaluations, such as microcanonical molecular dynamics (MD), are computationally expensive and primarily probe near-equilibrium states. To improve evaluation metrics for MLIPs, we introduce the Bond Smoothness Characterization Test (BSCT). This efficient benchmark probes the PES via controlled bond deformations and detects nonsmoothness, including discontinuities, artificial minima, and spurious forces, both near and far from equilibrium. We show that BSCT correlates strongly with MD stability while requiring a fraction of the cost of MD. To demonstrate how BSCT can guide iterative model design, we utilize an unconstrained Transformer backbone as a testbed, illustrating how refinements such as a new differentiable $k$-nearest neighbors algorithm and temperature-controlled attention reduce artifacts identified by our metric. By optimizing model design systematically based on BSCT, the resulting MLIP simultaneously achieves a low conventional E/F regression error, stable MD simulations, and robust atomistic property predictions. Our results establish BSCT as both a validation metric for practitioners to assess MLIP utility and as an "in-the-loop" model design proxy that alerts MLIP developers to physical challenges that cannot be efficiently evaluated by current MLIP benchmarks.

---

[*]Work done as undergraduate at Berkeley [1]California Institute of Technology, Pasadena, California, USA [2]University of California, Berkeley, Berkeley, California, USA [3]Lawrence Berkeley National Lab, Berkeley, California, USA. Correspondence to: Ryan Liu <ryanliu@caltech.edu>, Aditi S. Krishnapriyan <aditik1@berkeley.edu>.

*Proceedings of the 43rd International Conference on Machine Learning*, Seoul, South Korea. PMLR 306, 2026. Copyright 2026 by the author(s).

## 1. Introduction

Interatomic potentials are fundamental to computational chemistry and materials science, enabling essential tasks such as molecular dynamics (MD) and geometry optimization. Quantum mechanical calculations, such as Density Functional Theory (DFT) (Kohn & Sham, 1965) are widely used to derive these potentials, supporting applications from drug discovery to catalyst design (Cole & Hine, 2016; Jain et al., 2016; Hammer & Nørskov, 2000). However, DFT's $O(n^3)$ computational scaling makes it prohibitively expensive for large systems or long timescale simulations.

To overcome this bottleneck, Machine Learning Interatomic Potentials (MLIPs) have emerged as powerful surrogates, offering orders-of-magnitude speedups while approaching DFT accuracy. Typically, MLIPs are trained to minimize regression error on the energies and forces of atomistic structures, with a goal of reproducing the underlying potential energy surface (PES). However, even given a low test-set regression error, the curvature of the PES may not necessarily be correct. During dynamic simulations, this can manifest as unstable trajectories or nonphysical behavior (Fu et al., 2023; Bigi et al., 2025; Miret et al., 2025; Raja et al., 2025).

The true quantum mechanical PES has an inherent degree of smoothness that MLIPs are not necessarily guaranteed to capture. While the MLIP community often uses "smoothness" informally to mean bounded derivatives (Fu et al., 2025), rigorous chemical smoothness requires the absence of artificial extrema or inflection points (Subotnik et al., 2008). This property is especially crucial in far-from-equilibrium regimes, where data are more limited while spurious PES features can still cause simulation instability. However, current evaluations rely either on computationally expensive MD simulations or test sets limited to near-equilibrium states, which may not efficiently capture these instabilities (Póta et al., 2024; Fu et al., 2025).

To bridge this gap, we introduce the **Bond Smoothness Characterization Test** (BSCT), a benchmark that efficiently evaluates PES smoothness in both near- and far-from-equilibrium regimes. BSCT probes 1D bond deformations where the true PES is known to be smooth, such that artifacts created by MLIP can be easily isolated and detected. From

this, we define the **Force Smoothness Deviation** (FSD), a low-cost metric defined on BSCT that correlates strongly with MD stability, providing an early indicator of physical reliability.

We demonstrate BSCT's utility both as a validation metric and as a guide for architectural design. Using an expressive, unconstrained attention-based neural network as a test bed, we show how BSCT acts as an "in-the-loop" diagnostic to identify and reduce unphysical artifacts. Through targeted modifications, including a new differentiable $k$-nearest neighbors (Diff-kNN) algorithm, controllable Gaussian smearing, and temperature-controlled attention, we systematically improve the model's physical soundness. This case study establishes BSCT as a practical framework for evaluating and developing MLIPs, providing a critical signal of reliability that is not fully captured by standard MLIP benchmarks. The BSCT dataset and evaluation scripts are available on GitHub. The Diff-kNN algorithm is available in the Fairchem package.

## 2. Related Works

The development of reliable MLIPs has driven a dual focus in the community: the creation of rigorous benchmarks to validate physical consistency and the design of expressive architectures that can capture complex atomic interactions.

**MLIP Benchmarks.** Many MLIP benchmarks have been developed to supplement energy and force regression errors, and we list some examples of these here. The TorsionNet-500 dataset includes 500 molecules' torsion scan profiles, allowing MLIPs to compare their PES to DFT calculations (Rai et al., 2022). Fu et al. (2023) proposed to use MD simulation stability and $h(r)$ reconstruction to benchmark models. Bigi et al. (2025) used NVE simulations and Jacobians to measure the non-conservative behavior of the models. Kreiman & Krishnapriyan (2026) explicitly evaluated the generalization capabilities of foundational MLIPs. The NNP Arena provides an assortment of molecular and lattice property prediction benchmarks as well as inference speed test (Rowan Scientific Corporation, 2025). The Open Molecules 2025 (OMol25) dataset and leaderboard (Levine et al., 2025b) include molecular physics-based evaluations. Our benchmark evaluation, BSCT, differs from these by explicitly measuring PES smoothness far from equilibrium, and could be added to complement OMol25 evaluations.

For materials, the Open Catalyst project assesses models by their ability to predict the correct relaxed energy of the catalyst-adsorbate system (Chanussot et al., 2021). Matbench Discovery ranks models by their ability to predict structure stability (Riebesell et al., 2025). The MDR phonon benchmark tests MLIP's capability to predict the correct phonon structure of the lattice (Póta et al., 2024). MLIP

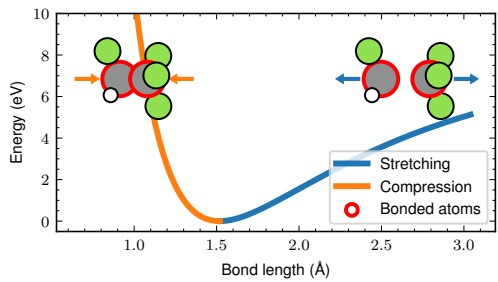

*Figure 1.* Example of how the Bond Smoothness and Characterization Test (BSCT) is constructed. We show a C-C bond in a $C_2H_2F_4$ molecule from the BSCT-SPICE dataset. The DFT reference PES smoothly varies across the wide range of bond lengths, showing the behavior expected from reliable interatomic potentials. These 1D probes are a simple and efficient way to measure PES smoothness in a regime of highly varying energies, which is inherently out-of-distribution of many MLIP training datasets.

arena provides a wide array of benchmarks for materials ranging from homonuclear diatomics to equations of state (Chiang et al., 2025). The JARVIS-Leaderboard has multiple benchmarks related to materials design (Choudhary et al., 2024).

**MLIPs for Atomistic Systems.** There is a long history of ML interatomic potentials, including those based on neural networks (Behler & Parrinello, 2007). Many of these neural networks have invariant features, i.e., featurizing embeddings to be invariant to group operations such as rotations and translations (Schütt et al., 2018; Zubatyuk et al., 2019; Haghighatlari et al., 2022; Zeng et al., 2023; Cheng, 2024; Smith et al., 2017; Chen & Ong, 2022; Pelaez et al., 2024; Yan et al., 2025; Zhang et al., 2026; Gasteiger et al., 2021). There are also several graph neural networks that build in symmetries such as rotational equivariance through featurizing via spherical harmonics (Batzner et al., 2022; Musaelian et al., 2023; Liao et al., 2024b; Fu et al., 2025; Kabylda et al., 2025; Batatia et al., 2022; Park et al., 2024; Bochkarev et al., 2024; Wood et al., 2026). There has also been increasing interest in models that do not explicitly build in all symmetries into the architecture, and instead focus on approaches to maximize learning these symmetries from the data (Pozdnyakov & Ceriotti, 2023; Qu & Krishnapriyan, 2024; Neumann et al., 2024; Rhodes et al., 2025; Mazitov et al., 2025). Kreiman et al. (2025) explored this direction further, demonstrating that unmodified Transformers can discover molecular structure without explicit graph priors.

## 3. Bond Smoothness Characterization Test

Evaluating an MLIP for physical modeling requires more than measuring its energy and force prediction errors. Reliable simulations with an MLIP depend on accurately reproducing the inherent smoothness of the true quantum

mechanical PES. In the quantum chemistry community, the *chemical smoothness* of a PES is defined as the absence of spurious discontinuities, extrema, and inflection points (Subotnik et al., 2008). This definition differs from the MLIP community's more recent use of "smoothness" to mean bounded PES derivatives (Fu et al., 2025), which is typically evaluated by costly microcanonical MD simulations.

**Near vs. Far-From-Equilibrium Smoothness.** A critical distinction exists between smoothness in the interpolation (near-equilibrium) and extrapolation (far-from-equilibrium) regimes. Near equilibrium, there is now enough training data that sufficiently expressive models can often improve smoothness by using this data (Yuan et al., 2026). For example, models on the Matbench Discovery leaderboard can achieve better near-equilibrium smoothness ($\kappa_{\mathrm{SRME}}$) by training the same architecture on larger datasets (Riebesell et al., 2025). As larger datasets such as OMol25 and OPoly26 (Levine et al., 2025b;a) become available, we believe that near-equilibrium smoothness will no longer be a constraining factor. However, while OMol25 and OPoly26 do include substantial reactive sampling, far-from-equilibrium regions of the PES remain less thoroughly sampled, making such configurations more often out-of-distribution (OOD). In this regime, physical reliability may also depend on the model's architectural priors. As shown in Yuan et al. (2024), MLIPs that are not explicitly trained on transition states data show degraded smoothness under bond breakage. BSCT targets this challenging OOD regime to expose instabilities that standard validations currently miss.

**What BSCT Measures.** The Bond Smoothness Characterization Test (BSCT) evaluates how smoothly an MLIP predicts energies and forces as molecular bonds are systematically stretched and compressed beyond equilibrium. We focus on bonds because their ground-truth PES (i.e., dissociation curves) is known to be smooth, making deviations and erroneous non-smoothness easier to detect. By sampling a one-dimensional slice of the PES along the bond length and comparing MLIP predictions with DFT references, BSCT isolates non-smooth behavior in regions with lower coverage in the training distribution. BSCT targets the challenging regime of far-from-equilibrium smoothness, enabling us to determine specific MLIP design choices that genuinely improve PES smoothness. This allows us to isolate the effect of architecture from raw model capacity, providing insight into how design choices influence the reliability and generalization of MLIPs.

**Constructing the BSCT Dataset.** For each molecule, we select a bond that splits the molecule into two fragments and displace the fragments along the bond axis while keeping their internal geometries fixed. Formally, given atomic positions $\{x_i\} \in \mathbb{R}^{N \times 3}$, the fragment labels $\{h_i\} \in \{-1, 1\}^N$, and the bond direction unit vector $\hat{r} \in S^2$, the perturbed positions for displacement $\alpha$ are:

$$x_i'(\alpha) = x_i + \alpha h_i \hat{r}. \tag{1}$$

We construct the **BSCT-SPICE dataset** by applying this procedure to the SPICE test structures (Eastman et al., 2023; Kovács et al., 2023). The dataset contains 485 molecules, each with 100 DFT single-point calculations computed at the same level of theory as SPICE ($\omega$B97M-D3(BJ)/def2-TZVPPD) using Psi4 (Mardirossian & Head-Gordon, 2016; Rappoport & Furche, 2010; Hellweg & Rappoport, 2015; Turney et al., 2012), the same computational chemistry code as SPICE.

We construct the dataset by systematically scanning bridge bonds in molecules, filtering out structures with isolated or overlapped atoms, and running DFT calculations on bond scans (See Appendix A for details). Figure 1 shows an example C–C bond scan, where the DFT PES varies smoothly across the sampled bond lengths; BSCT evaluates whether MLIPs preserve this physically correct behavior.

**Quantifying Smoothness: The FSD metric.** We introduce a new metric, the Force Smoothness Deviation (FSD), to quantitatively measure the PES smoothness. We define the "force norm due to perturbation" as:

$$\|\Delta \vec{F}\|^2 = \|\vec{F} - \vec{F}_{\mathrm{min}E}\|^2, \tag{2}$$

where $\vec{F}_{\mathrm{min}E}$ is the force vector at the minimum-energy structure in the 1-D PES section. As shown in Figure 2, the derivative of $\|\Delta \vec{F}\|^2$ with respect to the perturbed coordinate $\alpha$ sensitively detects non-smoothness (artificial minimum). Therefore, we define the force smoothness deviation (FSD) as:

$$\mathrm{FSD} = \max_{\alpha} \left| \frac{\frac{\mathrm{d}}{\mathrm{d}\alpha} \|\Delta \vec{F}_{\mathrm{MLIP}}\|^2}{\|\Delta \vec{F}_{\mathrm{MLIP}}\|^2} - \frac{\frac{\mathrm{d}}{\mathrm{d}\alpha} \|\Delta \vec{F}_{\mathrm{DFT}}\|^2}{\|\Delta \vec{F}_{\mathrm{DFT}}\|^2} \right| \tag{3}$$

$$= \max_{\alpha} \left| \frac{\mathrm{d}}{\mathrm{d}\alpha} \log \frac{\|\Delta \vec{F}_{\mathrm{MLIP}}\|^2}{\|\Delta \vec{F}_{\mathrm{DFT}}\|^2} \right|, \tag{4}$$

where $\alpha$ is defined by Equation 1, and the derivative is taken with respect to this one-dimensional perturbation parameter. We utilize the logarithmic derivative here because it captures the relative rate of change in the force norm, ensuring that smoothness artifacts are penalized equally in both high-force and low-force regions. A lower FSD indicates smoother and more physically sound PES predictions. The particular functional form makes FSD an indicator of chemical smoothness since the denominator $\|\Delta \vec{F}_{\mathrm{MLIP}}\|^2$ is small when an extremum is around, and the numerator $\frac{\mathrm{d}}{\mathrm{d}\alpha} \|\Delta \vec{F}_{\mathrm{MLIP}}\|^2$ is small when an inflection point is around.

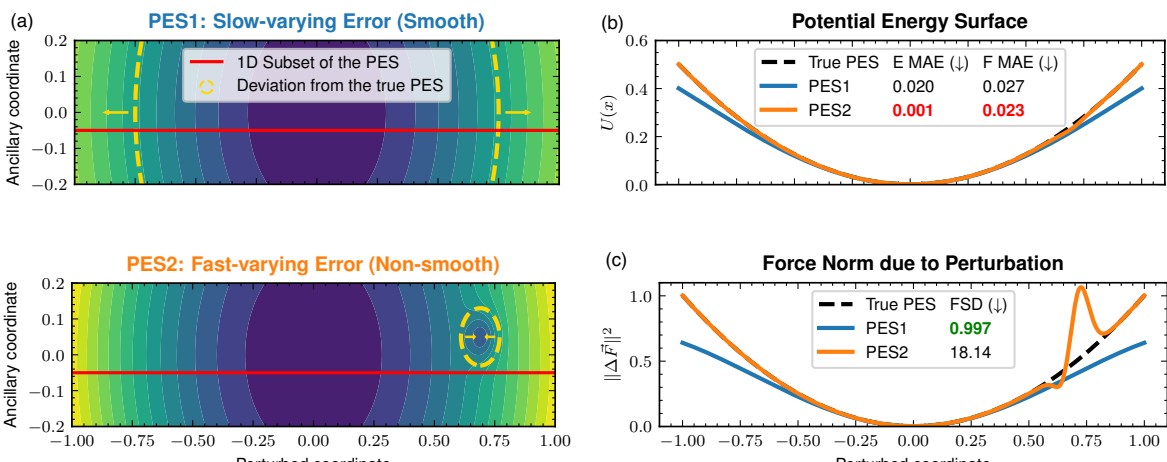

*Figure 2.* (a): Motivation for our proposed potential energy surface (PES) smoothness metric. We compare two synthetic, pedagogical PES examples designed to mathematically illustrate the metric's behavior. Both PES1 and PES2 accurately reproduce the true quadratic PES near equilibrium. Away from equilibrium, PES1 slowly deviates from the quadratic PES reference but remains smooth, and PES2 has an artificial minimum (non-smoothness) enclosed by gold dashed lines. (b) Standard metrics, such as energy and forces mean absolute errors (MAEs), evaluated on the one-dimensional subset fail to detect PES2's non-smoothness. (c) Our proposed force smoothness deviation (FSD) metric sensitively captures this non-smooth behavior.

By comparing the ratio of this to the DFT reference, FSD can detect any artificial extrema or inflection points and measure the chemical smoothness of a PES.

**Limitations.** We acknowledge that because BSCT relies on 1-D rigid fragment displacements, it serves as a necessary, rather than sufficient, condition for full high-dimensional PES smoothness. While it cleanly isolates far-from-equilibrium artifacts, a holistic evaluation should combine BSCT with benchmarks probing multi-directional perturbations, such as torsions and angular distortions. Meanwhile, although the logarithmic derivative in the FSD metric evaluates the relative rate of change (curve shape) to provide robustness against baseline shifts, we note as a limitation that the metric's sensitivity to mismatches in DFT calculation settings (e.g., evaluating an MLIP trained on a different functional or basis set than the BSCT reference) has not yet been systematically tested.

**Validating BSCT through "In-the-Loop" Model Design.** While we have motivated BSCT and FSD based on chemical principles, an important value lies in their ability to serve as actionable signals during model development. To demonstrate that BSCT is a viable proxy for expensive MD simulations, we must demonstrate that reducing FSD directly translates to improved physical stability. This requires a controlled experimental setup, and we start with an expressive, unconstrained baseline architecture to isolate specific sources of non-smoothness. In the following section, we introduce this backbone—not as an explicitly proposed state-of-the-art model, but as a neutral testbed designed to rigorously evaluate the diagnostic utility of BSCT in guiding iterative architectural design.

## 4. MinDScAIP: A Neural Network Testbed for Smoothness Evaluations with BSCT

To rigorously evaluate BSCT's sensitivity, we require an expressive backbone architecture with minimal geometric constraints. Starting from a neutral, unconstrained baseline architecture allows us to isolate specific sources of non-smoothness and ensures that any improvements captured by BSCT are directly attributable to our targeted design choices. We design the **Min**imally constrained **D**ifferentiable **Sc**aled **A**ttention **I**nteratomic **P**otential, **MinDScAIP**, to serve as this transparent testbed, enabling us to validate BSCT as a tool for iterative model development while keeping confounding architectural priors to a minimum.

### 4.1. Backbone Architecture Overview

The MinDScAIP backbone is structured to be similar to a Transformer backbone, and differs from conventional design mainly in the graph construction method and the attention mechanism employed. These designs are proposed to balance expressiveness and computational scalability, while keeping the architecture unconstrained enough to serve as a neutral testbed for testing BSCT-guided modifications. Full details of the model can be found in Appendix C.

**Graph Construction Methods.** Most attention-based MLIPs impose sparsity using radius graphs. However, this

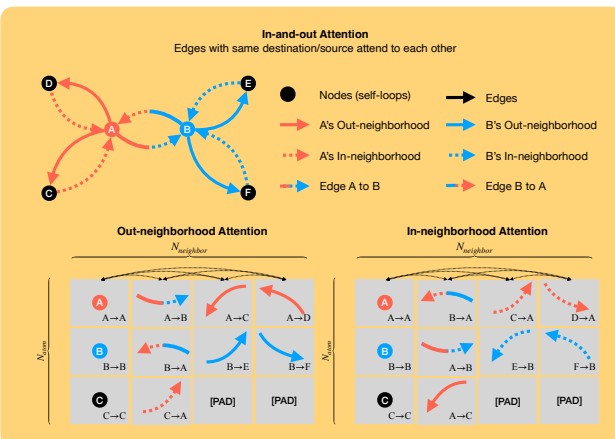

*Figure 3.* To study how different model design choices impact or improve PES smoothness, we design the backbone similar to Swin-Transformer, intending to create a neutral testbed for BSCT ablations. We generalize the shifted window attention of Swin Transformer to graphs by alternating in-and-out neighborhood attention. The interleaving windows allow information to propagate across the molecular graph.

often results in inefficient representations due to the highly variable number of neighbors per atom (see Appendix B for detailed analysis). For a more regular and efficient representation, the MinDScAIP backbone adopts a k-nearest-neighbor (kNN) graph structure. We arrange kNN edges into an $N_{atom} \times k$ array such that $(i, j)$-th element is the $j$-th shortest edge in node $i$'s neighborhood (Qu & Krishnapriyan, 2024). Essentially, rather than *padding* all neighborhoods to the same number of neighbors, we *truncate* the neighbors to obtain a regular representation. To make the truncation differentiable, we further propose a Differentiable k-NN Algorithm, which will be detailed in Section 4.2.

**Attention Mechanism.** MinDScAIP employs a dual attention strategy inspired by Swin-Transformer's shifted window attention (Liu et al., 2021), as depicted in Figure 3. Swin-Transformer partitions an image into windows (disjoint local sets). Similarly, on a directed graph, *neighborhoods* can partition edges into disjoint local sets. Similar to Swin-Transformer, where windows are shifted to create overlaps to propagate information, we can switch between in-neighborhood and out-neighborhood to achieve the same goal. As illustrated in Figure 3, MinDScAIP interleaves attention between edges sharing the same source atom (out-neighborhood) and edges sharing the same destination atom (in-neighborhood), generalizing Swin-Transformer to directed graphs.

## 4.2. Differentiable kNN Algorithm

A conservative force field can be important for physically consistent MD simulations, where atomic forces are defined as the negative gradient of the energy. While kNN graph construction offers computational advantages over standard radius graphs by assuring regular connectivity despite fluctuating atomic densities, standard kNN algorithms are inherently non-differentiable due to the truncation. To address this issue, we introduce the differentiable kNN algorithm (Diff-kNN), which ensures the potential remains differentiable, allowing for the prediction of conservative forces while maintaining the efficiency of a sparse graph. More recently, the conservative AllScAIP model incorporates the Diff-kNN algorithm. As of May 2026, it is currently the top-performing on the OMol25 leaderboard among energy-conserving MLIPs, as measured by multiple molecular physics-based evaluations (Qu et al., 2026; Levine et al., 2025b).

**Standard kNN Algorithm.** The standard kNN algorithm has two steps: first, a ranking is calculated for each edge $(i, j)$ with length $d_{ij}$ with respect to other edges in its neighborhood $\mathcal{N}(i) = \{(i, j')\}_{j'}$. This ranking is *hard*, as rankings can change discontinuously when edge lengths vary.

$$\text{rank}((i, j)|\mathcal{N}(i)) = \sum_{j'} \mathbb{I}(d_{ij} > d_{ij'}). \quad (5)$$

Second, it selects the $k$ shortest edges:

$$G = \{(i, j) : \text{rank}((i, j)|\mathcal{N}(i)) < k\} \quad (6)$$

Both the ranking and selection steps are non-differentiable.

**Soft Ranking with Diff-kNN.** To preserve differentiability of the kNN graph, we propose the differentiable kNN algorithm (Diff-kNN), which replaces the non-differentiable hard ranking with a differentiable soft ranking using a sigmoid function (see Figure 4):

$$\text{rank}((i, j)|\mathcal{N}(i)) = \sum_{j'} \sigma((d_{ij} - d_{ij'})/d_0), \quad (7)$$

where $d_0$ is a scale parameter that controls the sharpness of the sigmoid. This soft-ranking algorithm makes the ranking step differentiable [1] Edge selection is made differentiable using a smooth envelope function (Gasteiger et al., 2020; Pozdnyakov & Ceriotti, 2023), which assigns edge weights by:

$$e_{ij} = \exp(-f_{env}^2/(1 - f_{env}^2)), \quad (8)$$

where $f_{env} = \frac{\text{rank}((i,j)|\mathcal{N}(i))}{k}$, and $e_{ij}$ smoothly vanishes to zero when $f_{env} = 1 \Leftrightarrow \text{rank}((i, j)|\mathcal{N}(i)) = k$. The graph is therefore constructed by selecting all edges with $f_{env} < 1$. By biasing self-attention by the edge weights, the envelope function can ensure the smoothness of the selection step.

---

[1]It is memory inefficient when $N_{atom} \gg 1$. A memory-efficient version is described in Appendix D.

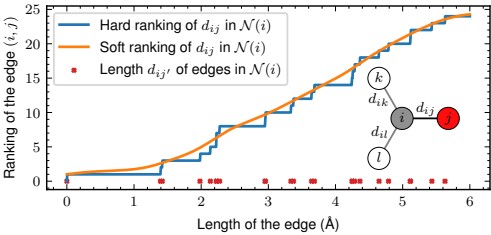

*Figure 4.* For the atomic graph construction, the Diff-kNN algorithm inherits the computational advantage of the $k$-NN algorithm while maintaining differentiability by replacing the hard ranking algorithm used in standard kNN with the soft ranking described in Equation 8. This allows the architecture to be fully differentiable, and so MLIP force predictions can be computed as the negative gradient of the potential energy.

**Combining Diff-kNN with a Radius Cutoff.**    To avoid unbounded edge lengths, which can produce long-tailed neighbor distributions, we combine Diff-kNN with a soft radius cutoff $r_c$. To incorporate this feature, we can redefine $f_{\text{env}}$ as:

$$f_{\text{env}} = \beta^{-1} \cdot \log(e^{\beta f_{\text{env, rank}}} + e^{\beta f_{\text{dist}}}), \tag{9}$$

where $f_{\text{env, rank}} = \text{rank}((i, j)|\mathcal{N}(i))/k$ and $f_{\text{env, dist}} = d_{ij}/r_c$. The log-sum-exponential is a soft approximation of the maximum function, and $\beta$ controls its smoothness.

## 5. Investigating BSCT-guided Model Design

Having established an expressive, unconstrained backbone (Section 4), we now use BSCT to investigate how it can be used to iteratively refine MLIP model design, i.e.: *Which targeted architectural modifications can improve potential energy surface (PES) smoothness while maintaining scalability and expressiveness?*

Our approach is to start from the neutral backbone MLIP, identify components that may introduce nonlinearities and propose designs to regularize them (Section 5.1), validate the effectiveness of BSCT (Section 5.2), introduce smoothness-oriented design choices independently (Section 5.3 & 5.4), and present the final results (Section 5.5). All hyperparameters can be found in Appendix I.

### 5.1. Identification of "Smoothness-Oriented" Model Design Choices

To establish the relationship between PES smoothness and specific model design choices, we first analyze the backbone architecture and locate components that introduce nonlinearity into the model's predictions. We propose targeted design choices to specifically regularize these nonlinearities, aiming to provide theoretical guarantees of PES smoothness. Each design choice is introduced independently and evaluated on its impact on three aspects of performance:

(i) accuracy near equilibrium (energy and force mean absolute errors, MAEs), (ii) smoothness (Force Smoothness Deviation, FSD metric measured with BSCT), and (iii) energy conservation in microcanonical molecular dynamics simulations.

**Sources of Nonlinearity.**    The MinDScAIP architecture consists of three main parts: featurization, attention blocks (self-attention & feedforward), and prediction heads (feedforward). From these, we identify three main sources of nonlinearities: (1) the Gaussian smearing featurization, (2) nonlinear activation functions, and (3) the softmax in scaled dot-product attention.

**Smoothness-oriented Design Modifications.**    We propose the following regularization strategies:

- **Controllable Gaussian Smearing**: Gaussian smearing featurizes atomic distances using Gaussian kernels (Schütt et al., 2018): $v_i = \exp(-\frac{|d-\mu_i|^2}{2\sigma^2})$, where $\sigma$ is set to be the spacing $\Delta x$ of $\mu_i$. We introduce a scaling factor $\gamma$, setting $\sigma = \gamma \Delta x$. Increasing $\sigma$ upper bounds the derivatives of any linear combination of $v_i$ relative to its infinity norm, thereby improving smoothness (see Appendix E).

- **Weight Decay**: Weight decay is a standard regularization technique that promotes NN smoothness. By regularizing the norm of the NN parameters, inputs to the activation function, such as SiLU, remain small and produce smoother transitions when the input structure is changed.

- **Temperature-controlled Attention**: We introduce a temperature parameter into the scaled dot-product attention mechanism: $\text{Attention}(Q, K, V; \tau) = \text{Softmax}\left(\frac{QK^T}{\tau\sqrt{E_k}}\right)V$. Larger $\tau$ values yield smoother attention outputs. Although temperature can be absorbed by scaling $Q$ and $K$, weight decay regularizes the magnitude of projection parameters, preventing arbitrary rescaling and making $\tau$ an effective smoothness control.

**In-the-loop Development of MLIPs with BSCT.**    The structuredness of BSCT allows researchers to develop MLIPs in an in-the-loop fashion. We provide an example of how the temperature-controlled attention was designed through direct inspection of MinDScAIP's behavior based on BSCT. When inspecting the energy and force curves of individual systems (unlike the synthetic curves in Fig. 2, these are real predictions for a $C_{11}H_{12}NO_2$ molecule), we noticed an unusual spike in the ratio of force norms, as shown in the upper Figure 5a. We directly inspected the components of MinDScAIP and eventually identified that

*Table 1.* A quantitative summary of Figure 6, showing BSCT's correlation to MD stability. We calculate the maximum change in kinetic temperature in 10fs of each trajectory shown in Figure 6, aggregated by averaging over all trajectories of the same temperature. The strong correlation observed serves as the basis of using BSCT as an early indicator of MD stability.

| Ablation | FSD (1/Å) | Max Jump 2000K | Max Jump 3000K | Max Jump 5000K |
|---|---|---|---|---|
| Vanilla | 97.4 | 9734 | 813 | 597 |
| Weight Decay | 76.3 | 1904 | 681 | 509 |
| Smear. & Temp. | 43.2 | 490 | 614 | 514 |

the attention changed rapidly around the spikes. To address this, we proposed temperature-controlled attention to regularize the attention smoothness explicitly. Figure 5b shows that improving attention score smoothness indeed improves the smoothness of MinDScAIP measured by BSCT.

### 5.2. BSCT as an early indicator of MD stability

The Force Smoothness Deviation (FSD) metric defined on BSCT is new to the community. To assess its physical relevance, we examine and show that FSD correlates with stability in far-from-equilibrium MD simulations.

**Problem Setup.** We select molecular structures from the MD22 dataset and relax them to their ground state using the MLIP. The system is equilibrated for 10 ps using a Langevin integrator with friction $1\mathrm{ps}^{-1}$. We then run high-temperature simulations, where bonds break in far-from-equilibrium geometries. We monitor the kinetic temperature of the system to detect any unrealistic jumps in kinetic energy. Sudden increases in kinetic temperature in a short period ($\gg T_{bath}$ within 10fs) are unlikely to originate from the heat bath and instead suggest spuriously large forces due to PES non-smoothness. We test three MinDScAIP models with the same architecture but varying strengths of smoothness-oriented designs, enabling us to isolate their correlation with FSD values. The study is repeated 10 times with different seeds, resulting in 70 distinct trajectories per model and temperature to increase the statistics.

**Results.** Figure 6 shows that higher FSD values (i.e., the more non-smoothness detected by BSCT) correlate strongly with more frequent and larger kinetic temperature spikes. A more quantitative measure is presented in Table 1, where we calculate the maximum change in kinetic temperature in 10fs and averaged over the 70 trajectories. This supports FSD as an early, low-cost predictive indicator of MD stability in far-from-equilibrium regimes: FSD computation takes $\sim 40$ minutes on one A6000 GPU, while MD simulations take $\sim 40$ hours.

### 5.3. Ablation Studies with BSCT Guidance

We are interested in understanding how the above design choices, along with the prediction head and model size, impact physical soundness. By combining our force smoothness deviation (FSD) metric on BSCT with energy/force MAEs and energy drift in NVE MD simulations, we have more comprehensive evaluations of MLIP physical consistency. Details of NVE MD energy drift experiments are in Appendix G. The ablations consider:

1. **Smoothness design choices**: Impact of weight decay, radial feature smearing widths, and temperature-controlled attention.

2. **Prediction head**: Direct-force, gradient-based forces (standard kNN), and gradient-based forces (Diff-kNN) with the large model (60M).

3. **Model size**: Gradient-based models with Diff-kNN graphs, varying from small (3.8M), medium (15M), to large (60M) sizes.

4. **Baselines to compare against**: MACE Large (4.7M), the baseline model trained on SPICE, and GemNet-T, which performed well on OMol25 (Levine et al., 2025b).

**Discussion.** Table 2 summarizes the evaluation results. The principal findings are:

1. **Smoothness Design Choices**: We observe that increasing smearing width smooths compressed regions where the single bond dominates the energy change, and thus its radial features are the most influential. Meanwhile, increasing attention temperature smooths stretched regions where multi-body interactions become critical as bonds break and form. Combining both yields the smoothest PES, confirming that nonlinearity must be regularized across both local and non-local features.

2. **Prediction Head**: Direct force regression yields better smoothness (low FSD on BSCT) by leveraging normalization layers, but inherently sacrifices energy conservation. In contrast, obtaining forces via automatic differentiation ensures a conservative potential but risks generating unbounded forces from backpropagation.

3. **Graph Construction**: The large NVE energy drift confirms the non-conservative nature of standard kNN graphs, and Diff-kNN can restore conservative behavior while retaining computational advantage.

4. **Model Size**: We found that larger models improve accuracy but degrade far-from-equilibrium smoothness,

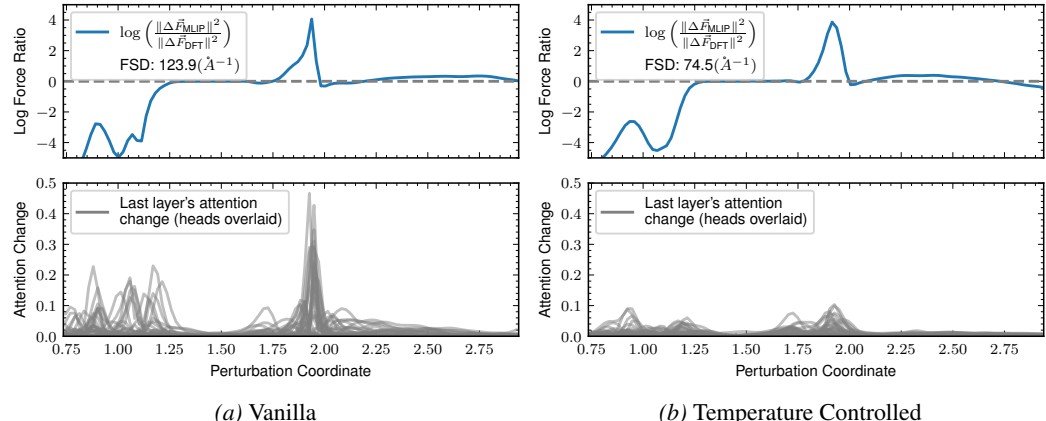

*(a)* Vanilla          *(b)* Temperature Controlled

*Figure 5.* Example of how BSCT can serve as an in-the-loop evaluation for MLIP development by detecting real-world artifacts in a trained MLIP. We probe a $C_{11}H_{12}NO_2$ molecule in the BSCT-SPICE dataset, visualizing $\log\left(\|\Delta\vec{F}_{\text{MLIP}}\|^2/\|\Delta\vec{F}_{\text{DFT}}\|^2\right)$, whose derivative with respect to $\alpha$ defines FSD. We also visualize the changes in the attention scores from the stretched N-C bond to the N atom along the bond scan, with heads overlaid. The strong correlation in FSD and attention score suggests the need for explicit regularization, motivating the proposed temperature-controlled attention.

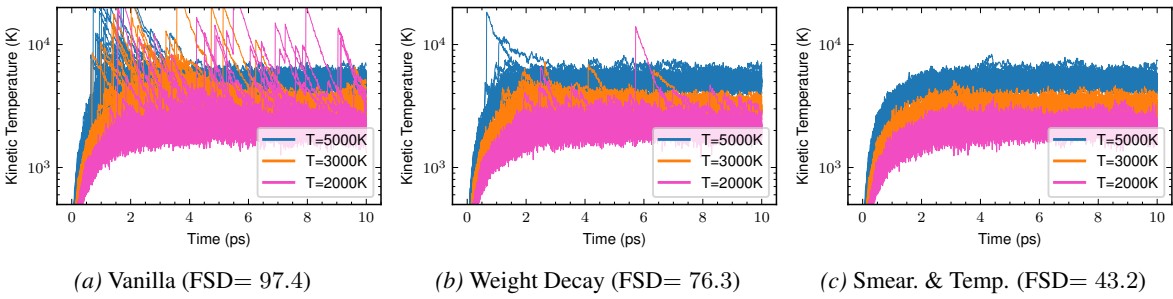

*(a)* Vanilla (FSD= 97.4)     *(b)* Weight Decay (FSD= 76.3)     *(c)* Smear. & Temp. (FSD= 43.2)

*Figure 6.* We present empirical evidence of BSCT's validity as a low-cost proxy to MD stability. We visualize the kinetic temperature of 70 canonical ensemble simulation trajectories (seven structures from MD22, each with ten random seeds), and repeat over three different temperatures. We observe a strong correlation between the FSD score calculated on BSCT and the number of instabilities (spikes in kinetic temperature).

consistent with the intuition that additional nonlinearities can cause abrupt PES changes. Therefore, regularization is essential for scaling MLIPs to maintain smoothness.

5. **Baselines**: MinDScAIP with smoothness designs outperforms MACE and GemNet-T in near-equilibrium accuracy while matching their FSD scores on BSCT.

### 5.4. Near-Equilibrium Smoothness: MPTrj Ablations

We assess whether smoothness-oriented designs benefit near-equilibrium behavior using the MPTrj dataset (Deng et al., 2023) and the Matbench Discovery benchmark (Riebesell et al., 2025), as the smoothness design choices are not specific to far-from-equilibrium systems and are also applicable to improve near-equilibrium smoothness.

**Problem Setup.** We train three versions of MinDScAIP-30M on the MPTrj dataset, with weak, moderate, and strong

smoothness-oriented designs. The MLIPs trained on MPTrj are evaluated on the Matbench Discovery benchmark (Riebesell et al., 2025), which tests model capability to relax materials to their ground state geometry (RMSD), correctly predict their stability (F1), and capture phonon modes ($\kappa_{\text{SRME}}$) (Póta et al., 2024). We focus on the $\kappa_{\text{SRME}}$ metric, which reflects the smoothness near equilibrium.

**Results.** Table 3 shows that stronger smoothness designs yield modest F1 improvements but substantial $\kappa_{\text{SRME}}$ reductions. Since $\kappa_{\text{SRME}}$ requires calculating a dense grid of force sets, measuring FSD on BSCT is a fast proxy to evaluate such smoothness during model development.

### 5.5. Accuracy Benchmarks

Given the insights from the BSCT and MPTrj ablations, we present the standard accuracy benchmarks of MinDScAIP with smoothness-oriented designs in Table 4 and 5. MinDScAIP achieves strong E/F regression error on

*Table 2.* Ablations on SPICE and BSCT, showing that BSCT-guided MinDScAIP can simultaneously achieve strong accuracy on standard energy and forces errors and competitive physical soundness. The best model is boldfaced, while the best per group is underlined. For energy drift, since small values for conservative models correspond to numerical errors, the best model is not boldfaced, and non-conservative ones are colored in red.

| Ablation | | Test MAE (↓) | | NVE Sim. | FSD (↓) | | |
| Group | Model | Energy (meV/atom) | Forces (meV/Å) | Energy Drift (meV/atom) | Full (1/Å) | Compress (1/Å) | Stretch (1/Å) |
|---|---|---|---|---|---|---|---|
| Smoothness Design Choices | Weight decay | 0.09 | 2.94 | 0.631 | 76.3 | 65.2 | 55.1 |
| | Smearing | 0.10 | **2.86** | 0.832 | 83.1 | 32.3 | 82.5 |
| | Temperature | 0.10 | 3.06 | 0.675 | 75.5 | 63.3 | 62 |
| | Smear. & Temp. | 0.12 | 2.94 | 0.708 | 43.2 | 32.8 | 38.1 |
| Prediction Head | Direct Force | 0.15 | 4.22 | 2.6e5 | 71.8 | 69.4 | 42.8 |
| | Gradient kNN | **0.08** | 3.01 | 19.10 | 105 | 80.7 | 93.1 |
| | Gradient Diff-kNN | 0.09 | 3.02 | 0.678 | 97.4 | 69.5 | 87.2 |
| Model Size | Small | 0.23 | 6.63 | 0.640 | 80.2 | 61.3 | 66.2 |
| | Medium | 0.12 | 4.01 | 0.722 | 93.2 | 75.4 | 79.8 |
| | Large | 0.09 | 3.02 | 0.678 | 97.4 | 69.5 | 87.2 |
| Baseline | MACE | 0.79 | 14.3 | 0.691 | 62.1 | 62.1 | **12.3** |
| | GemNet-T | 0.30 | 7.11 | 110.6 | **33.8** | **28.8** | 20.5 |

*Table 3.* Ablations of the MinDScAIP architecture on MPTrj, showing that the improvement from BSCT-guidance is transferable to near-equilibrium smoothness of material systems. We present the three MLIPs with weak, moderate, and strong smoothness designs along with the specific hyperparameters and the three primary metrics of Matbench Discovery (F1, $\kappa_{SRME}$, and RMSD).

| Model | Weight Decay | Temperature | Smearing Width | F1 ↑ | $\kappa_{SRME}$ ↓ | RMSD ↓ |
|---|---|---|---|---|---|---|
| Weak | $1 \times 10^{-3}$ | 1 | 1 | 0.807 | 0.77 | 0.092 |
| Moderate | $1 \times 10^{-2}$ | 5 | 5 | 0.811 | 0.63 | 0.089 |
| Strong | $5 \times 10^{-2}$ | 10 | 10 | 0.817 | 0.49 | 0.088 |

SPICE MACE-OFF, as well as strong F1 score & competitive $\kappa_{SRME}$ on Matbench Discovery (see Appendix H for the inference efficiency benchmark).

### 5.6. Final Remarks

We emphasize that BSCT alone does not constitute a holistic evaluation of the PES smoothness. The value of BSCT is its utility as both an additional MLIP evaluation metric and as an in-the-loop model evaluation to complement standard energy and force errors. Once a model is refined using BSCT, we strongly recommend a full evaluation using multiple benchmarks (e.g., energy conservation test, thermal conductivity calculation, etc.) to verify final performance. Future work could include constructing a BSCT benchmark for the OMol25 dataset, which is the largest and most diverse molecular dataset to date, enabling models to improve further on downstream tasks.

## 6. Conclusion

We introduce the Bond Smoothness Characterization Test (BSCT) as a targeted, low-cost diagnostic of potential energy surface (PES) smoothness predicted by machine-learned interatomic potentials (MLIPs), enabling early detection of near- and far-from-equilibrium non-smoothness, as evidenced by correlations with MD simulations and thermal conductivity calculations. Furthermore, BSCT successfully predicts the stability of existing foundational MLIPs like MACE and GemNet-T, proving its value beyond our specific testbed. Through a principled architecture investigation guided by BSCT, we identified that smoothness must be enforced across both local (smearing) and non-local (attention) scales. We successfully translated these insights into concrete architectural modifications and improved MLIP PES smoothness, while preserving scalability and expressiveness. More broadly, BSCT demonstrates how physics-motivated evaluation metrics can directly inform model design, providing a practical framework for developing MLIPs that combine accuracy, scalability, and physical soundness.

## Acknowledgement

R.L. is partially supported by the National Science Foundation Graduate Research Fellowship Program under Grant No. 2139433, the DOE SC-HEP #SC0019219 and #SC0011925 Caltech HEP/Quantum grants, and the Research Corpora-

tion RESCSCIAD.25011 quantum grant. Any opinions, findings, and conclusions or recommendations expressed in this material are those of the author(s) and do not necessarily reflect the views of the National Science Foundation. This work was also supported by Scialog grant #SA-AUT-2024-015b from Research Corporation for Science Advancement and Arnold and Mabel Beckman Foundation, and Toyota Research Institute as part of the Synthesis Advanced Research Challenge. S.M.B. acknowledges support from the Center for High Precision Patterning Science (CHiPPS), an Energy Frontier Research Center funded by the U.S. Department of Energy, Office of Science, Basic Energy Sciences at Lawrence Berkeley National Laboratory under Contract No. DE-AC02-34205CH11231.

## Impact Statement

This paper presents work whose goal is to advance the field of machine learning. There are many potential societal consequences of our work, none of which we feel must be specifically highlighted here.

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

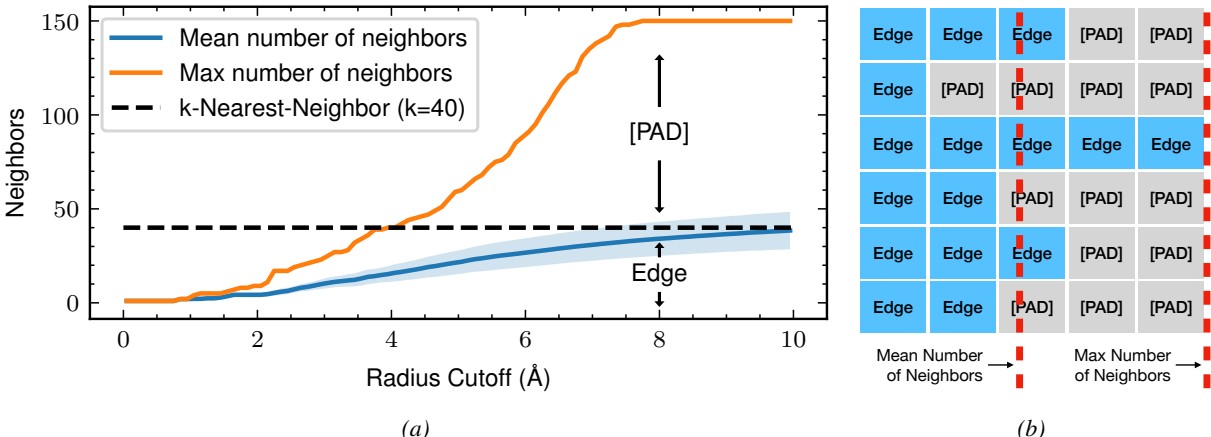

*Figure 7.* Challenges with using a k-Nearest-Neighbor graph construction. (a): Mean (± standard deviation) and maximum number of neighbors as a function of radius cutoff in a radius graph. To use a dense implementation of attention, one must pad all sequences to the same length, which is the maximum number of neighbors. (b) Illustration of padding inefficiency due to the long-tailed distribution of neighbor counts. Since the padding also requires computation, ideally, we want to minimize the gap between the two red dashed lines.

## A. BSCT Sampling Procedure

The BSCT-SPICE dataset is constructed with the following procedure.

1. Identify all candidate bridge bonds that partition the molecule into two separate fragments without creating isolated atoms.

2. Linearly stretch and compress candidate bonds, covering bond lengths from $0.5\times$ to $2\times$ the sum of the bonded atoms' covalent radii.

3. Exclude candidate bonds if, upon perturbation, any pair of atoms (except the pair bonded by the candidate) becomes closer than $0.9\times$ the sum of the bonded atoms' covalent radii.

4. Sample from the filtered dataset across selected bond types (C–C, C–N, C–O, C–P, C–S, N–N, N–O, N–P, and O–P).

5. Run DFT on 100 evenly spaced structures along each selected bond perturbation trajectory.

6. Exclude bonds with discontinuous PES due to Self-Consistent Field convergence issues.

Since the sampling process is random, we counteracted the arbitrariness by controlling the bond types that we sample, such as C-C, C-O, etc., since bond types are the most indicative property of a bond. We also set the bond lengths of the bridge bond independently by the sum of the covalent radii of the bonded atoms instead of using the bond length in the original structure.

## B. Graph Construction Methods

Graph construction methods have a critical impact on the topology of the graphs. In most settings, MLIPs use a radius graph to impose locality constraints and implement periodic boundary conditions. However, the distribution of the number of neighbors is usually very long-tailed. In Figure 7a, the number of neighbors as a function of radius cutoff is calculated for the SPICE MACE-OFF dataset. The maximum number of neighbors increases significantly faster than the average case. As discussed in the main text, if we want to use dense attention kernel, we must pad the sequences to the same length, the maximum number of neighbors. Figure 7b shows an illustration of an array with a large padding rate. The majority of the computational power is spent on the padding tokens, which do not influence the results. Using k-Nearest-Neighbors to limit the maximum number of neighbors can close the gap between the mean and the maximum number of neighbors, which is highly preferable when using dense attention kernels.

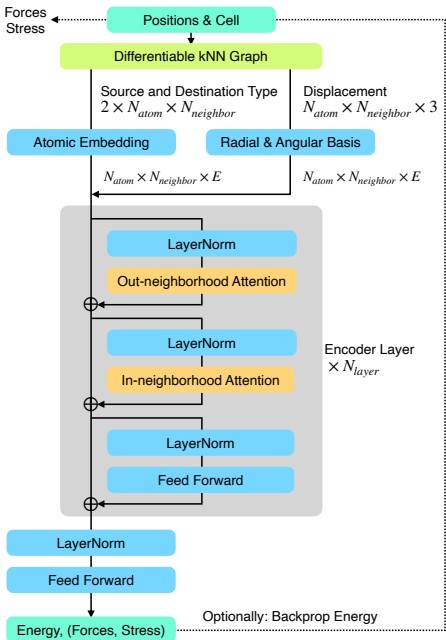

*Figure 8.* Summarization of the MinDScAIP architecture. The structural similarity to the Transformer architecture is an intentional design that aims to facilitate the application of many technologies developed for large-scale training in modern machine learning.

## C. MinDScAIP Architecture

The MinDScAIP backbone, summarized in Figure 8, consists of three main components: (1) featurization and graph construction, (2) attention blocks, and (3) prediction heads. Each component is chosen to balance expressiveness and computational scalability, while keeping the architecture unconstrained enough to serve as a neutral platform for testing BSCT-guided modifications

1. **Featurization and Graph Construction**: To keep the backbone efficient, we use minimal feature engineering and rely on data augmentation to learn rotational equivariance. Edge embeddings are constructed by summing atomic embeddings of source and destination atoms, and expanding displacement vectors using radial and angular basis functions. While MinDScAIP is compatible with any set of basis functions, we choose Gaussian smearing for the radial basis and spherical harmonics for the angular basis. For graph construction, we use the Diff-kNN algorithm introduced in Section 4.2 or the standard kNN algorithm.

2. **Attention Blocks**: The attention mechanism used by MinDScAIP is introduced in Section C. The edge attention blocks are applied with skip-connection and pre-normalization, followed by a pre-normed 2-layer feedforward network. We also apply a self-gating layer to the output of edge attention blocks, as suggested in (Jumper et al., 2021).

3. **Predicting Heads**: MinDScAIP predicts per-atom energies with a feed-forward network applied to node embeddings (self-loops) and aggregates them to get system-level energy predictions. Forces and stresses are either predicted with a feed-forward network in the direct-force mode or obtained by deriving the energy predictions in the gradient-based forces mode.

## D. Memory Efficient Diff-kNN

To make the Diff-kNN algorithm discussed in Section 4.2 memory efficient, we observe that the soft ranking function,

$$\text{rank}((i,j)|\mathcal{N}_{out}(i)) = \sum_{j'} \sigma((d_{ij} - d_{ij'})/d_0), \tag{10}$$

where $\sigma(x)$ is the sigmoid function, $d_{ij}$ are the lengths of the edges, and $d_0$ controls the sharpness, requires $O(N_{\text{atoms}}^2)$ space for each calculation since each term $\sigma((d_{ij} - d_{ij'})/d_0)$ contribute non-zero values to the sum. To save space, we would like to truncate the summation. To achieve this, we can replace $\sigma(x) = \frac{1}{1+e^{-x}}$ with the bump function $g(x)$, whose derivative has compact support on $[-1, 1]$:

$$g(x) = \begin{cases} 0 & \text{if } x < -1 \\ \frac{e^{-2/(x+1)}}{e^{-2/(x+1)} + e^{-2/(x-1)}} & \text{if } x \in [-1, 1] \\ 1 & \text{if } x > 1 \end{cases} \tag{11}$$

where $g(x) \approx x$ when $|x| \ll 1$ and $g(x) \equiv \mathbb{I}(x)$ when $|x| > 1$. Since $g(x)$ is strictly zero when $x < -1$, we can sort edge lengths and truncate at some rank $k + \Delta$ before taking the summation, since the edges with large displacement will contribute 0 to the sum when calculating the envelope factor for the edges with small displacement that will be included in the graph. This can reduce the memory complexity of the gradient graph from $O(N_{\text{atoms}}^3)$ to $O(N_{\text{atoms}}(k + \Delta)^2 + N_{\text{atoms}}^2)$.

## E. Gaussian Smearing

In this section, we look at Gaussian smearing, as described in Section 5.1. We show that increasing $\sigma$ upper bounds the derivatives of any linear combination of these basis functions relative to their infinity norm. We consider the derivative of an arbitrary linear combination of basis functions $v_i(x) = e^{-(x-i)^2/2\sigma^2}$ specified by coefficients $\{a_i\}$: $f(x) = \sum_i a_i v_i(x)$, where the basis function has width $\sigma$. Now we are interested in the maximum derivative at some location $y$ normalized by the infinity norm of such a linear combination. Without loss of generality, we can assume $y = 0$.

$$\max_{\{a_i\}} \frac{\frac{\partial}{\partial y} \sum_{i=-\infty}^{\infty} a_i e^{-(y-i)^2/2\sigma^2} \Big|_{y=0}}{\left\| \sum_{i=-\infty}^{\infty} a_i e^{-(x-i)^2/2\sigma^2} \right\|_\infty} = \max_{\{a_i\}} \frac{\frac{1}{\sigma^2} \sum_{i=-\infty}^{\infty} i a_i e^{-i^2/2\sigma^2}}{\left\| \sum_{i=-\infty}^{\infty} a_i e^{-(x-i)^2/2\sigma^2} \right\|_\infty} \tag{12}$$

$$\approx \frac{1}{\sigma^2} \max_{a(\cdot)} \frac{\int_{-\infty}^{\infty} a(z) x e^{-z^2/2\sigma^2} \mathrm{d}z}{\left\| \int_{-\infty}^{\infty} a(z) e^{-(x-z)^2/2\sigma^2} \mathrm{d}z \right\|_\infty} \tag{13}$$

$$= \frac{1}{\sigma} \max_{a(\cdot)} \frac{\int_{-\infty}^{\infty} a(\sigma z) z e^{-z^2/2} \mathrm{d}z}{\left\| \int_{-\infty}^{\infty} a(\sigma z) e^{-(x-z)^2/2} \mathrm{d}z \right\|_\infty} \tag{14}$$

$$\propto \frac{1}{\sigma} \tag{15}$$

For the first equality, the numerator is derived and evaluated at $x = 0$. After that, the summation is approximated by an integral, and the coefficients become a continuous function. Subsequently, we change the variable of integration. The RHS has no dependence on $\sigma$ other than the $\sigma^{-1}$ factor since $a(\sigma z)$ dependence can be absorbed by the maximization over $a(\cdot)$. We conclude that this quantity is upper bounded by $O(\sigma^{-1})$. Therefore, the larger the smearing width, the more bounded the derivatives are.

## F. Conventional Accuracy Benchmarks on SPICE MACE-OFF and Matbench Discovery

**Problem setup.** In this section, we present the conventional energy and forces regression error results on the held-out SPICE MACE-OFF (Eastman et al., 2023; Batatia et al., 2022) test dataset and the benchmark results on Matbench Discovery (Riebesell et al., 2025). The evaluated MinDScAIP model is trained on MPTrj (Deng et al., 2023); specifically, we use the smear. & temp. model that achieved the best FSD scores in our BSCT ablations. For SPICE MACE-OFF, we follow the convention to report per-atom energy MAE and forces MAE aggregated per dataset. For Matbench Discovery, we report the evaluation metrics obtained using the PyPI `matbench-discovery` package.

**Results.** The SPICE MACE-OFF results are presented in Table 5, and the Matbench Discovery results are presented in Table 6. MinDScAIP with smoothness designs guided by BSCT achieves competitive accuracy on both leaderboards while maintaining physical soundness, demonstrating BSCT's effectiveness as an in-the-loop evaluation tool for model development. We highlight that although the raw numbers reported are no longer the top ones as of 2026, the primary contribution of this work is our benchmark and evaluation-guided model development process. We anticipate that designing new MLIPs to combine both the insights from BSCT and more recent MLIP advances may lead to even better performance.

*Table 4.* MinDScAIP Smear. & Temp. shows strong accuracy on the SPICE test set, binned by the datasets of molecules. The errors are reported in per-atom energy MAE (meV/atom) and forces MAE (meV/Å).

| | MACE 4.7M | | EScAIP 45M | | eSEN 6.5M | | MinDScAIP 60M | |
| --- | --- | --- | --- | --- | --- | --- | --- | --- |
| Dataset | **E** | **F** | **E** | **F** | **E** | **F** | **E** | **F** |
| PubChem | 0.88 | 14.75 | 0.53 | 5.86 | 0.15 | 4.21 | **0.14** | **3.38** |
| DES370K M. | 0.59 | 6.58 | 0.41 | 3.48 | 0.13 | 1.24 | **0.06** | **0.99** |
| DES370K D. | 0.54 | 6.62 | 0.38 | 2.18 | 0.15 | 2.12 | **0.09** | **0.90** |
| Dipeptides | 0.42 | 10.19 | 0.31 | 5.12 | 0.25 | 3.68 | **0.09** | **1.48** |
| Solvated A.A. | 0.98 | 19.43 | 0.61 | 11.52 | 0.25 | **3.68** | **0.13** | 3.96 |
| Water | 0.83 | 13.57 | 0.72 | 10.31 | 0.15 | 2.50 | **0.13** | **2.29** |
| QMugs | 0.45 | 16.93 | 0.41 | 8.74 | **0.12** | 3.78 | 0.16 | **2.86** |

*Table 5.* Evaluation on the Matbench Discovery benchmark, showing MinDScAIP's expressiveness and physical-soundness allows it to achieve strong F1 accuracy while maintaining good $\kappa_{\text{SRME}}$. Models are sorted in F1 order, with the exception that our model (MinDScAIP-60M) is placed at the top. (Data reflects the leaderboard standing as of May 6, 2026.)

| Model | F1 ↑ | DAF ↑ | Precision ↑ | Accuracy ↑ | MAE ↓ | R2 ↑ | $\kappa_{\text{SRME}}$ ↓ | RMSD ↓ | CPS ↑ |
| --- | --- | --- | --- | --- | --- | --- | --- | --- | --- |
| MinDScAIP-60M | 0.833 | 5.313 | 0.812 | 0.948 | 0.035 | 0.789 | 0.691 | 0.086 | 0.722 |
| EquiformerV3+DeNS-MP | 0.863 | 5.479 | 0.838 | 0.956 | 0.029 | 0.840 | 0.275 | 0.070 | 0.830 |
| MatRIS-10M-MP | 0.847 | 5.422 | 0.829 | 0.951 | 0.031 | 0.824 | 0.489 | 0.072 | 0.778 |
| eSEN-30M-MP | 0.831 | 5.260 | 0.804 | 0.946 | 0.033 | 0.822 | 0.340 | 0.075 | 0.797 |
| eqV2 S DeNS | 0.815 | 5.042 | 0.771 | 0.941 | 0.036 | 0.788 | 1.676 | 0.076 | 0.522 |
| DPA3-v2-MP | 0.803 | 5.024 | 0.768 | 0.936 | 0.037 | 0.812 | 0.650 | 0.080 | 0.718 |
| Eqnorm MPtrj | 0.786 | 4.844 | 0.741 | 0.929 | 0.040 | 0.799 | 0.408 | 0.084 | 0.756 |
| ORB v2 MPtrj | 0.765 | 4.702 | 0.719 | 0.922 | 0.045 | 0.756 | 1.725 | 0.101 | 0.470 |
| Nequip-MP-L | 0.761 | 4.704 | 0.719 | 0.921 | 0.043 | 0.791 | 0.452 | 0.086 | 0.733 |
| SevenNet-l3i5 | 0.760 | 4.629 | 0.708 | 0.920 | 0.044 | 0.776 | 0.550 | 0.085 | 0.714 |
| GRACE-2L-MPtrj | 0.691 | 4.163 | 0.636 | 0.896 | 0.052 | 0.741 | 0.525 | 0.089 | 0.681 |
| MACE-MP-0 | 0.669 | 3.777 | 0.577 | 0.878 | 0.057 | 0.697 | 0.647 | 0.091 | 0.644 |

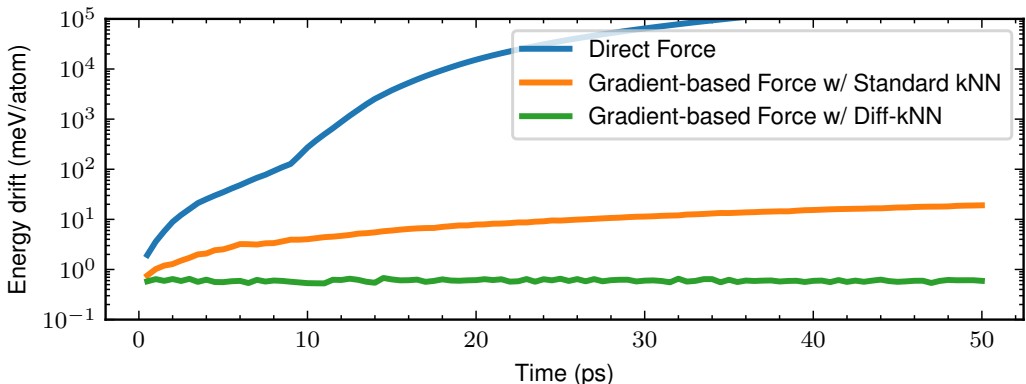

*Figure 9.* The energy drift of three different MinDScAIP models during microcanonical (NVE) ensembles averaged over seven MD22 molecular trajectories. The direct-force and non-differentiable kNN gradient models (blue and orange curves) show significant energy drift due to non-conservative or discontinuous predictions. The gradient-based Diff-kNN model employs the Diff-kNN algorithm described in Section 4.2 and conserves the energy (the non-zero drift is due to the first-step error of the Verlet integrator).

## G. Energy Conservation Tests in Molecular Dynamics (MD)

**Problem Setup.** We assess the differentiability and smoothness of our models by performing MD simulations under the microcanonical (NVE) ensemble with the Verlocity Verlet integrator (Larsen et al., 2017). The degree of non-conservative behavior is upper bounded by the magnitude of the higher derivatives of the PES (Hairer et al., 2003). Thus, even if a model is infinitely differentiable, it can still manifest non-conservative behavior if the higher derivatives are not bounded. The "smoothness" here is defined as the boundedness of PES derivatives, which is different from the force smoothness deviation defined by BSCT. We follow the protocol proposed by (Fu et al., 2025) and use the seven molecules in the MD22 dataset (Chmiela et al., 2023) as out-of-distribution test systems for the MD simulation data relative to SPICE training. Each simulation integrates dynamics for 100ps with 1fs time steps.

**Results.** Figure 9 compares energy drift among MinDScAIP models with different prediction heads and graph structures. The direct force MinDScAIP model cannot conserve energy due to its inherently non-conservative nature. The gradient-based force MinDScAIP model with a standard non-differentiable kNN graph cannot conserve energy due to the piecewise continuity of the standard kNN graph, which results in unbounded first and higher-order derivatives. In contrast, the gradient-based force MinDScAIP model with a differentiable kNN graph can conserve energy. It is infinitely differentiable, and this also demonstrates that its higher derivatives are bounded.

## H. Inference Efficiency Benchmarks

**Benchmark Setup.** We follow the benchmark described in Fu et al. (2025) to test the throughput and memory usage of MinDScAIP, MACE-MP-16M, and eSEN-30M-OAM (smaller eSEN model weights are not public) on the diamond system. We vary the number of supercells included in the image to test model's throughput and memory usage as a function of number of atoms. All benchmark is done on a single 80GB A100.

**Results** Table 6 summarizes the inference benchmark results. We highlight that MinDScAIP-60M is substantially outperforms MACE-MP-16M in material stability predictions (Matbench F1) by including more parameters (4x the size of MACE-MP-16M), while being just slightly slower than MACE-MP-16M. Comparing against eSEN-30M-OAM, MinDScAIP offers comparable accuracy while being order-of-magnitude faster and memory-efficient than eSEN. Both results suggests that MinDScAIP is more scalable than the current architectures.

## I. Hyperparameters

The hyperparameters used for the experiments are summarized in Table 7 For the MPTrj experiments, we adopt the training procedure proposed in (Fu et al., 2025) and models are pretrained with direct-force and DeNS targets (Liao et al., 2024a), then fine-tuned with a gradient-based prediction head following Fu et al. (2025). The parameters for pretraining are specified

*Table 6.* Inference efficiency benchmark of three selected models. Models are tested on the diamond system with varying number of supercells. The benchmark is done on a single 80GB A100. MinDScAIP promises accuracy similar to eSEN while being just slightly slower than MACE-MP.

| Number of Atoms | Throughput (Millions of Steps Per Day) | | | Memory Usage (GB) | | |
|---|---|---|---|---|---|---|
| | MinDScAIP-60M | MACE-MP-16M | eSEN-30M-OAM | MinDScAIP-60M | MACE-MP-16M | eSEN-30M-OAM |
| 216 | 1.15 | 1.49 | 0.09 | 4.32 | 2.3 | 35.81 |
| 512 | 0.51 | 0.72 | OOM | 9.83 | 5.19 | OOM |
| 1000 | 0.26 | 0.37 | OOM | 19.7 | 10.0 | OOM |
| 1728 | 0.15 | 0.23 | OOM | 35.35 | 17.16 | OOM |
| 2744 | 0.09 | 0.15 | OOM | 59.57 | 27.06 | OOM |

*Table 7.* The hyperparameters used for MinDScAIP experiments. The parameters used for pretraining are indicated by parentheses: direct force prediction is enabled during pretraining (✓in the parentheses) and gradient-based prediction is used for fine-tuning (✗that follows the parentheses).

| dataset | SPICE MACE-OFF | | | | | | | | | | MPTrj-ablations | | | |
|---|---|---|---|---|---|---|---|---|---|---|---|---|---|---|
| hyperparameter | Direct-Force | Grad. kNN | Grad. Diff-kNN | Small | Medium | Large | W. Decay | Smearing | Temperature | Smear. & Temp. | Weak | Medium | Strong | final-60M |
| Embedding dimension | 512 | 512 | 512 | 128 | 256 | 512 | 512 | 512 | 512 | 512 | 256 | 256 | 256 | 512 |
| Hidden factor | 2 | 2 | 2 | 2 | 2 | 2 | 2 | 2 | 2 | 2 | 2 | 2 | 2 | 2 |
| Number of layers | 8 | 8 | 8 | 8 | 8 | 8 | 8 | 8 | 8 | 8 | 8 | 8 | 8 | 8 |
| Number of heads | 32 | 32 | 32 | 8 | 16 | 32 | 32 | 32 | 32 | 32 | 16 | 16 | 16 | 32 |
| kNN k | 30 | 30 | 30 | 30 | 30 | 30 | 30 | 30 | 30 | 30 | 30 | 30 | 30 | 30 |
| Diff-kNN | ✗ | ✗ | ✓ | ✓ | ✓ | ✓ | ✓ | ✓ | ✓ | ✓ | ✓ | ✓ | ✓ | ✓ |
| Diff-kNN $d_0$ | N/A | N/A | 0.2Å | 0.2Å | 0.2Å | 0.2Å | 0.2Å | 0.2Å | 0.2Å | 0.2Å | 0.2Å | 0.2Å | 0.2Å | 0.2Å |
| Diff-kNN $\beta$ | N/A | N/A | 10 | 10 | 10 | 10 | 10 | 10 | 10 | 10 | 10 | 10 | 10 | 10 |
| Mem-eff. Diff-kNN $\Delta$ | N/A | N/A | N/A | N/A | N/A | N/A | N/A | N/A | N/A | N/A | 20 | 20 | 20 | 20 |
| Radius cutoff | 6Å | 6Å | 6Å | 6Å | 6Å | 6Å | 6Å | 6Å | 6Å | 6Å | 6Å | 6Å | 6Å | 6Å |
| Number of radial basis | 128 | 128 | 128 | 128 | 128 | 128 | 128 | 128 | 128 | 128 | 128 | 128 | 128 | 128 |
| Smearing Scale | 1 | 1 | 1 | 1 | 1 | 1 | 1 | 5 | 1 | 5 | 1 | 5 | 10 | 10 |
| Angular $l_{max}$ | 5 | 5 | 5 | 5 | 5 | 5 | 5 | 5 | 5 | 5 | 5 | 5 | 5 | 5 |
| Temperature $\tau$ | 1 | 1 | 1 | 1 | 1 | 1 | 1 | 1 | 10 | 10 | 1 | 5 | 10 | 10 |
| Direct-Force Prediction | ✓ | ✗ | ✗ | ✗ | ✗ | ✗ | ✗ | ✗ | ✗ | ✗ | (✓) ✗ | (✓) ✗ | (✓) ✗ | (✓) ✗ |
| Batch size | 128 | 128 | 128 | 128 | 128 | 128 | 128 | 128 | 128 | 128 | 256 | 256 | 256 | (128)256 |
| Weight decay | $10^{-3}$ | $10^{-3}$ | $10^{-3}$ | $10^{-3}$ | $10^{-3}$ | $10^{-3}$ | $5 \times 10^{-2}$ | $5 \times 10^{-2}$ | $5 \times 10^{-2}$ | $5 \times 10^{-2}$ | $10^{-3}$ | $10^{-2}$ | $5 \times 10^{-2}$ | $10^{-2}$ |
| Warmup factor | 0.2 | 0.2 | 0.2 | 0.2 | 0.2 | 0.2 | 0.2 | 0.2 | 0.2 | 0.2 | (0.2) 0. | (0.2) 0. | (0.2) 0. | (0.2) 0. |
| Warmup epochs | 1 | 1 | 1 | 1 | 1 | 1 | 1 | 1 | 1 | 1 | 1 | 1 | 1 | 1 |
| Number of Epochs | 100 | 100 | 100 | 100 | 100 | 100 | 100 | 100 | 100 | 100 | (60) 40 | (60) 40 | (60) 40 | (60) 40 |
| EMA decay | 0.999 | 0.999 | 0.999 | 0.999 | 0.999 | 0.999 | 0.999 | 0.999 | 0.999 | 0.999 | 0.999 | 0.999 | 0.999 | 0.999 |
| Gradient Clipping | 100 | 100 | 100 | 100 | 100 | 100 | 100 | 100 | 100 | 100 | 100 | 100 | 100 | 100 |
| Energy Loss Weight | 1 | 1 | 1 | 1 | 1 | 1 | 1 | 1 | 1 | 1 | 1 | 1 | 1 | 1 |
| Forces Loss Weight | 2 | 2 | 2 | 2 | 2 | 2 | 2 | 2 | 2 | 2 | (2)4 | (2)4 | (2)4 | (2)4 |
| Stress Loss Weight | N/A | N/A | N/A | N/A | N/A | N/A | N/A | N/A | N/A | N/A | 10 | 10 | 10 | 10 |

by the parentheses.

# J. Ethics Statement

The authors uphold the code of ethics and are committed to maintaining trustworthiness and transparency in their scientific practices. Although this work is expected to have limited direct social impact, we recognize the possibility of its misuse. We therefore urge readers to exercise careful judgment when applying, deploying, or releasing any products or artifacts derived from our work.

# K. Reproducibility Statement

The authors are committed to ensuring the reproducibility of this work. All essential technical details are provided in the main text and appendix. The dataset is available in the BSCT repository, and the Diff-kNN algorithm is available in the Fairchem repository. In addition, the code, configurations, datasets, and selected checkpoints are available upon request to the correspondence authors.

