# OpenReview forum: "From Evaluation to Design: Using Potential Energy Surface Smoothness Metrics to Guide ML Interatomic Potential Architectures"
_ICML.cc/2026/Conference — ICML 2026 regular_

### Official Review · Reviewer_kbWM · 2026-02-18

**Soundness:** 2
**Presentation:** 2
**Significance:** 2
**Originality:** 3
**Overall Recommendation:** 4
**Confidence:** 3

**Summary:**

The manuscript describes i) a novel metric, the force smoothness deviation (FSD), based on scans of molecule fragments, which ii) are presented in a dataset called BSCT, and iii) a different MLIP architecture.

FSD is intended as a replacement to running long microcanonical MD simulations to judge the smoothness of the PES, and this is demonstrate empirically by showing exemplary that for some systems, for very high temperatures, the value of FSD appears to correlate to kinetic energy spikes.

Additionally, the value of FSD is used to guide the design of a novel MLIP architecture. In the same instance, also a differentiable k-nn algorithm is proposed.

**Compliance With Llm Reviewing Policy:**

Affirmed.

**Key Questions For Authors:**

Why is a differentiable k-nn algorithm needed?

**Limitations:**

Yes.

**Strengths And Weaknesses:**

Strength:
- I believe the novel FSD metric is interesting, and could be used as a metric for evaluating novel MLIP models.


Weaknesses:
- The message is quite diluted by introducing (more or less) unrelated aspects (novel MLIP architecture, novel metric, and a dataset). I would find it far more convincing if the novel metric would have been evaluated on established state-of-the-art architectures showing (also quantitatively) a correlation with some kind of stability measure. Now, as it stands, it is very unclear if this brings any actual advantage.

---

> ### Author Rebuttal · Authors · 2026-03-30
>
> We thank the reviewer for reviewing our manuscript and for recognizing that the FSD metric is an interesting and potentially valuable tool for evaluating MLIP models. Below, we address the reviewer's concerns regarding the focus of the paper, the evaluation of established architectures, and the necessity of the differentiable k-NN (Diff-kNN) algorithm.
>
> **The Focus of the Paper & Evaluation of SOTA Architectures.** We understand the reviewer's concern that introducing a metric, a dataset, and an architecture might feel broad. However, our intention is to demonstrate how BSCT/FSD can serve as an actionable, in-the-loop diagnostic tool to actively guide architectural design, rather than just acting as a static benchmark.
>
> To illustrate this "evaluation-to-design" pipeline, we utilized the vanilla MinDScAIP architecture as an intentionally unconstrained testbed. If we had tried to apply these iterative regularizations to heavily engineered, established SOTA models, the strong geometric constraints and numerous existing model components would act as confounding factors, making it challenging to cleanly isolate how specific interventions (e.g., temperature-controlled attention and smearing widths) address smoothness.
>
> Regarding the evaluation of established SOTA architectures, we did benchmark MACE and GemNet-T using our metric (Table 2), and the correlation with MD stability holds (as we are clarifying for Reviewer 3):
> - **GemNet-T** achieves a low FSD (33.8) on our benchmark, which perfectly correlates with the highly stable trajectories we observe when running it in high-temperature MD simulations.
> - **MACE**, while highly accurate near equilibrium, is revealed by BSCT to struggle with bond compression smoothness (yielding a high FSD). This high FSD accurately predicts severe kinetic energy spikes (instability) that we observe when running MACE in high-temperature MD.
>
> | name | FSD (1/Å) | Max Jump 2000K | Max Jump 3000K | Max Jump 5000K |
> |-|-|-|-|-|
> | MindScAIP w/ Smear. & Temp. | 43.2 | 490 | 614 | 514 |
> | GemNet-T | 33.8 | 789 | 692 | 2203 |
> | MACE-OFF Medium | 86.4 | 437 | 42825 | Inf |
>
> We will revise the introduction and conclusion to better clarify this connection between the proposed metric and our architectural testbed, emphasizing that BSCT successfully predicts the stability of established models as well.
>
> **Why is a Differentiable k-NN (Diff-kNN) Algorithm Needed?** The Diff-kNN algorithm is strictly necessary to solve a fundamental trade-off between computational efficiency and physics preservation (energy conservation) in MLIP graph construction:
> - **The Inefficiency of Radius Graphs:** Most standard MLIPs use a fixed radius cutoff. Because atomic density fluctuates, the number of neighbors per atom varies widely. To process these graphs efficiently on GPUs (especially with dense attention mechanisms), all neighborhoods must be padded to the maximum number of neighbors, resulting in massive memory and computational waste (Figure 7).
> - **The Physics Violation of Standard k-NN:** Using a standard k-Nearest-Neighbors algorithm fixes the neighborhood size, solving the padding inefficiency. However, standard k-NN involves a "hard" sorting and truncation step. This makes the potential energy surface (PES) mathematically discontinuous. When atoms move during Molecular Dynamics (MD), neighbors swap in and out of the top-k list abruptly. This leads to undefined forces at the boundary, breaking energy conservation and causing massive energy drift in microcanonical (NVE) simulations (as shown by the orange line in Figure 9).
> - **The Diff-kNN Solution:** Our Differentiable k-NN algorithm introduces a soft, differentiable ranking and envelope function. This retains the fixed-size computational efficiency of standard k-NN, but guarantees that the PES remains infinitely differentiable. As a result, the forces remain conservative, allowing the model to perfectly conserve energy during NVE MD simulations (the green line in Figure 9).
>
> We view Diff-kNN as a broadly applicable, modular contribution that can help developers accelerate their models while maintaining strict physical constraints. Demonstrating its transferability, concurrent research by Qu et al. [1] recently applied Diff-kNN to achieve state-of-the-art results on the OMol25 benchmark. We will explicitly highlight this physical motivation in Section 4.2 of the revised manuscript.
>
> [1] Qu, Eric, et al. "A recipe for scalable attention-based MLIPs: unlocking long-range accuracy with all-to-all node attention." arXiv preprint arXiv:2603.06567 (2026).

---

> > ### Author Rebuttal · Reviewer_kbWM · 2026-04-01
> >
> > The answers have resolved my concerns.

---

### Official Review · Reviewer_6ND6 · 2026-03-04

**Soundness:** 3
**Presentation:** 2
**Significance:** 2
**Originality:** 3
**Overall Recommendation:** 4
**Confidence:** 3

**Summary:**

The papers introduces the Band Smoothness Characterization Test, a benchmark to analyze the potential energy surface (PES). The benchmark is derived by stretching and compressing 1D bonds and can thereby directly measure measure PES smoothness close or far from equilibrium. To study the benchmark, a new metric called Force Smoothness Deviation (FSD) and an unconstrained and fully differentiable new architecture named  MinDScAIP are proposed as well. Both serve to quantify which parts of MLIP design choices affect PES smoothness. A series of experiments then justify related, further adaptations.

**Compliance With Llm Reviewing Policy:**

Affirmed.

**Final Justification:**

Thank you to the authors, the AC, SAC and PC.

Based on the paper and rebuttal, my current assessment is as follows:

The presented paper is technically very solid, it has a solid range of experiments demonstrating usefulness but some limitations in terms of scope beyond MD simulation.

The authors did a good job explaining some of the shortcomings and confusions of the paper with their rebuttal and following discussion. Namely, they addressed why MACE-OFF presented good FSD in some results but cannot be used in high temperature regimes, demonstrating that their BSCT-guided design is superior accross several different modes. Furthermore, they expanded on the differences between EScAIP and MinDScAIP that were not immediately obvious, clearing up why their model is more unconstrained. And lastly, they were able to provide explanations for why they decided not to include additional stability tests and how they plan to expand on the benchmark in the future.

 I believe the responses have demonstrated that BSCT is a robust measure of the PES and that it is useful in some domains, namely MD,  and have therefore raised my score.

However, a stronger positive score would require practical use in several.

**Key Questions For Authors:**

- In Figure 6, do the MD spikes and FSD correlation hold for other (generally smooth) models like MACE?
- If the argument for MindSCAIPs performance is its unconstrained nature, why do you think MACE performs relatively well on the FSD metric?
- What benefits compared to the original attention architecture of EscAIP does the new version hold?
- What is the next step in developing are more nuanced, realistic BSCT benchmark?

**Limitations:**

The authors do acknowledge that there are limitations of the BSCT benchmark, but extending this to some specific failure modes would have been useful for the reader.

**Strengths And Weaknesses:**

## Strengths ##
The paper is well structured and provides sound mathematical descriptions.

The BSCT is an interesting benchmark and the authors present ways of optimizing their models based on the FSD metric in a cohesive manner.

Based on the results, the improved models show better MD simulation behavior and phonon absorption response, demonstrating that the in-loop design choices lead to better modeling outcomes.

## Weaknesses ##

The main limitation is the scope of the proposed benchmark. BSCT is presented as detecting physical smoothness issues “both near and far from equilibrium,” but the benchmark itself is fundamentally based on filtered 1D bond deformation trajectories in molecular systems. This is a much narrower setting than a general quantification of PES smoothness. To the authors credit, they do acknowledge that BSCT is not a holistic evaluation metric, but this limitation also lowers the impact and limits the strength of the BSCT benchmark.

It is therefore not clear that the BSCT benchmark constitutes a generally useful test for measuring the smoothness of a potential energy surface, and it is unclear that it would affect how you would train these in practice. The benchmark is too simple for realistic settings where bond angles and/or material related properties will affect the FSD metric’s accuracy, such as angular distortions and torsions. Additional structure modifications during the BSCT benchmark creation could have alleviated this somewhat. At present, it is a 1D bond-stretch/compression benchmark evaluated only on molecules

Another issue is that the authors introduce the MinDScAIP architecture by stating the need for a minimally constrained testbed for evaluation. From the description of the attention mechanism, it is not clear, however, why this specific style of attention is less constrained than other models using regular attention, such as EScAIP.
On the same point, there is a lack of comparison to the original EScAIP model in terms of performance, making it unclear what the advantages of the new design are.

Furthermore, the MD simulation test of Figure 6 displays the improvement of your model to smooth over instabilities, but no comparison is made to any common models like MACE. Do the FSD spikes hold for other models?

A minor issue is that the figures are generally too small. This is worst for figure 3 which is almost too small to read.

---

> ### Author Rebuttal · Authors · 2026-03-30
>
> We thank the reviewer for their detailed feedback. Below, we address the reviewer's questions and raised weaknesses.
>
> **Scope of BSCT, its 1D Nature, and Failure Modes.** We agree that 1-D bond stretching is narrower than characterizing a high-dimensional PES. However, this intentional design makes BSCT an effective diagnostic tool rather than a physical pathway simulation.
>
> Evaluating MLIPs on 1-D bond dissociation benefits from a strict quantum chemistry guarantee: the ground-truth potential energy curve must be mathematically smooth and contain slow-varying derivatives. Probing far-from-equilibrium spaces with realistic reaction pathways introduces complex, rapidly varying derivatives into the ground-truth PES, making it difficult to distinguish true physical variations from MLIP artifacts. By probing a guaranteed smooth regime, BSCT cleanly isolates out-of-distribution failure modes.
>
> We agree with the reviewer that angular distortions and torsions can be a more physically realistic scenario. There are readily available evaluations for MLIPs for torsion scans and angular distortions. We emphasize that BSCT serves as a necessary, rather than sufficient, condition for high-dimensional PES smoothness, and it should be used in conjunction with other benchmarks for a holistic evaluation. While angular distortions and torsions can detect certain failure modes, they cannot detect far-from-equilibrium non-smoothness that BSCT is designed to detect. Therefore, they should be viewed as complementary parts instead of competing solutions.
>
> **MD Spikes, FSD Correlation, and MACE.** We ran MD simulations for GemNet-T and MACE-OFF Medium (we were not able to include MACE-OFF large due to computational constraints).
>
> The results confirm the correlation between FSD and MD stability. Contrary to the assumption that MACE is perfectly smooth, BSCT reveals a high FSD (86.4) due to poor smoothness during bond compression. This correctly predicts MACE's severe instability in high-temperature MD, marked by massive kinetic energy jumps. Conversely, GemNet-T has a lower FSD and exhibits more stable simulations.
>
> | name | FSD (1/Å) | Max Jump 2000K | Max Jump 3000K | Max Jump 5000K |
> |-|-|-|-|-|
> | MindScAIP w/ Smear. & Temp. | 43.2 | 490 | 614 | 514 |
> | GemNet-T | 33.8 | 789 | 692 | 2203 |
> | MACE-OFF Medium | 86.4 | 437 | 42825 | Inf |
>
> (Note: While the rough correlation holds, GemNet-T and MACE show different trends than MinDScAIP when temperature increases. This is likely due to the discrepancy in their prediction of dynamics far-from-equilibrium. This is why we argue the MD spike and FSD correlation in a controlled setting where the architecture is held fixed and only smoothness regularizations vary.)
>
> **Clarifying MACE's FSD vs. MinDScAIP's "Unconstrained" Nature.** The reviewer asked why MACE performs well on FSD if MinDScAIP benefits from being unconstrained. We clarify two points:
> - **Unconstrained architectures boost accuracy but hurt FSD:** MinDScAIP’s unconstrained nature drives its state-of-the-art regression accuracy but inherently degrades smoothness (seen in the Vanilla baseline). This trade-off motivated us to use BSCT to design targeted regularizations (temperature and smearing) that restore smoothness while preserving expressiveness.
> - **MACE's FSD is mixed:** MACE's overall FSD is worse than GemNet-T and our regularized MinDScAIP due to its poor compression FSD. While MACE achieves excellent FSD on the stretching end due to its constrained message-passing mechanism and smooth envelope cutoffs, these place little constraint on the compression end (where the envelope is ~1), which is likely responsible for the poor compression FSD. This highlights BSCT’s utility in catching blind spots even in heavily engineered models.
>
> **Benefits Compared to EScAIP & Performance Comparison.** A direct performance comparison to EScAIP is in Table 5. MinDScAIP significantly outperforms EScAIP across the SPICE test sets (e.g., on PubChem, EScAIP 45M yields 0.53 meV/atom Energy MAE and 5.86 meV/Å Force MAE, whereas MinDScAIP 60M achieves 0.14 and 3.38).
>
> Architecturally, EScAIP relies on hand-crafted Bond Orientational Order (BOO) parameters for edge featurization and complex node aggregation. MinDScAIP's attention mechanism can be viewed as EScAIP's attention mechanism with node-aggregation replaced by another attention block (with neighborhoods flipped). Since the edges are not aggregated after one attention calculation, they can be updated in a learnable fashion, which removes the necessity of BOO featurization for edges. This relaxation makes MinDScAIP more unconstrained and promises higher expressivity.
>
> **Next Steps for a More Nuanced Benchmark.** We plan to extend the BSCT framework to organometallic complexes and clusters (e.g., OMol25 dataset), where highly delocalized bonding networks make multi-dimensional coordinates essential.
>
> **Minor Issue**: We will increase the sizes of the figures in the revised manuscript.

---

> > ### Author Rebuttal · Reviewer_6ND6 · 2026-04-01
> >
> > Thank you for your response. Your response answered some of my points very well, and I will increase the score.
> >
> > Your additional MACE and GemNet experiments clarify why MACE does well in some experiments and fails under compression. Further, your additional comments on the MinDScAIP architecture provide the needed insight into your motivation.
> >
> > However, I remain unconvinced regarding the broader usefulness of BSCT. I appreciate your clarification that BSCT is intended as a necessary but not sufficient condition, and your indication that you plan to extend the benchmark to more complex systems (e.g., organometallic complexes and clusters).
> > That said, it is still unclear how BSCT translates into a practical evaluation tool beyond bond-stretching regimes. In particular:
> >
> > You note that there are evaluations for torsion scans and angular distortions, but you do not include any tests that evaluate whether a good FSD metric on bond stretching can extend to these other regimes. Have you conducted these experiments? Or why did you not do so?
> >
> > You mention that you plan on extending the framework to multi-dimensional coordinates. Could you provide a more concrete description of how you might do so? Can this extension be done in the same framework with the FSD metric?
> > Clarifying these points would strengthen the case that BSCT in its current state is a useful evaluation tool, and that it is the first step in a new PES smoothness evaluation research direction.

---

> > > ### Author Response · Authors · 2026-04-02
> > >
> > > **Whether a good FSD metric on bond stretching can extend to these other regimes.** We agree that additional tests, such as TorsionNet-500 and the Biaryl Torsion benchmark used in MACE-OFF [2], would strengthen our work by more comprehensively evaluating MinDScAIP and the baseline models. However, we did not include them primarily because they probe an orthogonal property of the potential energy surface: energy prediction accuracy rather than derivative smoothness.
> > >
> > > The diverse set of MLIP evaluations each probes a different configuration space of the high-dimensional PES. While showing that our metric correlates well with another evaluation sounds appealing, they may actually probe orthogonal properties. For example, the most common metric in torsion scans is barrier height error, which focuses on energy prediction accuracy rather than directly evaluating PES smoothness. If our primary goal were simply to build the most accurate and physically-reliable MLIP, studying torsion scans in addition to our current set of evaluations would make sense. However, our main objective is to demonstrate the utility of BSCT in guiding MLIP design.
> > > Therefore, we instead examined correlations with MD stability and Matbench Discovery's thermal conductivity. Both tasks have a more explicit dependence on PES smoothness. MD stability has significant overlap with BSCT (both probe far-from-equilibrium smoothness), while thermal conductivity is more distinct (probing near-equilibrium smoothness of solid-state materials). Together, these experiments support our argument that BSCT accurately measures PES smoothness and generalizes to more complex systems. Because torsion scans do not provide this specific utility, we prioritized our finite bandwidth for evaluations that directly correlate with smoothness.
> > >
> > > **Extending the framework to multi-dimensional coordinates.** First, we apologize for the phrasing in our previous response. Stating that delocalized bonds make "multi-dimensional coordinates essential" was too strongly worded and was an artifact from an earlier draft of the rebuttal. A more precise statement is that for complex systems (like organometallic clusters), "we must go beyond naive molecular graphs." (as in response to reviewer FeVe)
> > >
> > > To answer your question concretely, the FSD framework can naturally be extended to multi-dimensional perturbations. We can rewrite the FSD definition by taking the gradient with respect to a multi-dimensional perturbation vector $\vec{\alpha}$:
> > >
> > > $$\mathrm{FSD}=\max_{\vec{\alpha}}\left\Vert\nabla_{\vec{\alpha}} \log\frac{\Vert \Delta \vec F_{\text{MLIP}}\Vert^2}{\Vert \Delta \vec F_{\text{DFT}}\Vert^2}\right\Vert_2$$
> > >
> > > The corresponding atomic displacement becomes a joint perturbation:
> > >
> > > $$x_{i}'(\vec{\alpha}) = x_i+\sum_{j=1}^D\alpha_j h_{ij}\hat r_j$$
> > >
> > > where $\vec{\alpha}$ is a $D$-dimensional vector, $h_{ij}$ is the $j$-th partition label according to the $j$-th bond subject to the test, and $\hat{r}_j$ defines the $j$-th bond's direction. Intuitively, it's the Cartesian product of two BSCT tests in the 1D case.
> > >
> > > By defining FSD this way, the framework remains intact for multi-dimensional coordinates. The open research questions we are currently investigating for this extension are: (1) how should "bonds" (and therefore the partition labels $h_{ij}$) be systematically defined in the absence of a clear molecular graph (e.g., highly delocalized electrons), and (2) when does evaluating the joint-perturbation of multiple bonds provide a diagnostic advantage over evaluating marginal 1-D probes independently. As the work is still in progress, we cannot provide definitive answers to these questions. We hope this clarifies how BSCT serves as the first step in a broader PES smoothness evaluation research direction.
> > >
> > > [2] Kovács, Dávid Péter, et al. "MACE-OFF: Transferable short range machine learning force fields for organic molecules." arXiv preprint arXiv:2312.15211 (2023).

---

### Official Review · Reviewer_2a2d · 2026-03-09

**Soundness:** 3
**Presentation:** 4
**Significance:** 3
**Originality:** 3
**Overall Recommendation:** 4
**Confidence:** 4

**Summary:**

The paper "From Evaluation to Design: Using Potential Energy Surface Smoothness Metrics to Guide MLIP Architectures" addresses a recurring issue in MLIP architecture design, which is the extrapolation performance of the models and the smoothness of the potential energy surface. The smoothness of the surface has critical effects on molecular dynamics simulations which are one of the main targets of MLIP application. The authors introduce two new metrics called Bond Smoothness Characterization Test and Force Smoothness deviation and a new benchmarking dataset BSCT-SPICE. With the user of the metrics and this new dataset, the authors try to evaluate the quality of MLIP potentials and guide the design of a new MLIP architecture called MinDScAIP.

**Compliance With Llm Reviewing Policy:**

Affirmed.

**Key Questions For Authors:**

The authors should mention or discuss the transferability of the results of the paper. How centric are they to the specific architecture of MinDSCAIP and Transformer-based models, and do they believe the results extend also to architectures such as message passing networks of equivariant graph networks?

Also, since they demonstrate that increasing model size reduces smoothness, it would be interesting to know if the double-descent phenomenon also applies to MLIP models, and further increasing model sizes would give improved performance.

In addition to the synthetic test shown in Figure 2, it would be useful to include demonstrations of real artifacts found in current MLIPs.

**Limitations:**

yes

**Strengths And Weaknesses:**

The paper is technically sound, the demonstrated results make sense and provide a great testbed for the design of the MinDSCAIP model and ideas on guiding the development of new MLIPs.

The concept of bond-stretching tests in MLIPs is not new, it has already been used for evaluating the PES smoothness in the past.

The problem addressed in the paper is relevant. The dataset and metrics provided in the paper will provide a useful testbed for future MLIPs and can become a useful addition to the already large arena of MLIP benchmarks.

While the idea is not original, the evaluation metrics and datasets are, and they should prove useful. In the end, a bond stretching/compression is a 1D probing of a many-body potential. As thus it's limited in the amount of smoothness it can evaluate. However, it is the bare minimum (as in a necessary but not sufficient condition) which should indeed be tested in the design of MLIPs.
Additionally, some of the results shown on the MinDSCAIP model will very probably not extend to other MLIP architectures. The MLIP field contains a wide variety of very different architectures, which probably have very different weaknesses in how they can produce a smooth PES.

---

> ### Author Rebuttal · Authors · 2026-03-30
>
> We thank the reviewer for their constructive feedback. Below, we address the reviewer’s specific questions regarding the limitations of 1D probing, transferability, model scaling, and the demonstration of real artifacts.
>
> **Limitations of 1D Probing.** We agree that because BSCT operates on 1-D slices, it serves as a necessary, rather than sufficient, condition for high-dimensional PES smoothness, and it should be used in conjunction with other benchmarks (like phonon dispersion or energy conservation tests) for a holistic evaluation. Nevertheless, BSCT is a valuable addition to the evaluation landscape because no other benchmarks explicitly test the far-from-equilibrium smoothness of MLIPs in a way that serves as a low-cost, early indicator of MD instability. We also highlight that this is a limitation shared by all other existing MLIP evaluations. We will make the acknowledgement of this limitation more explicit in the revised manuscripts.
>
> **Transferability to Other Architectures.** The contributions of this paper span both evaluation methodologies and architectural interventions, both of which offer high transferability:
>
> - **Metric Transferability**: BSCT and the FSD metric are entirely architecture-agnostic. We demonstrated this by evaluating non-Transformer models like MACE (an equivariant message-passing network) and GemNet-T (an invariant graph network), as shown in Table 2.
>
> - **Design Transferability**: Several of our architectural insights extend directly to other models. For instance, controlling the Gaussian smearing width and applying weight decay are regularization strategies that benefit general MPNNs. Furthermore, our Differentiable k-NN (Diff-kNN) algorithm is broadly applicable and can seamlessly replace discontinuous radius or standard k-NN graphs in existing MPNNs to restore energy conservation. Demonstrating its modular utility, concurrent research by Qu et al. [1] recently applied Diff-kNN to achieve state-of-the-art results on the OMol25 benchmark. Finally, while "temperature-controlled attention" is specific to Transformer-based models, the higher-level takeaway—that non-local interaction mechanisms require explicit regularization far from equilibrium—applies directly to regularizing message-passing aggregation functions in equivariant networks.
>
> **Model Scaling and the Double-Descent Phenomenon.** Based on our empirical observations, we do not see a "double descent" phenomenon regarding PES smoothness. In our ablations (Table 2), we observed that naively increasing the model size of the vanilla baseline from Small (3.8M) to Large (60M) improved near-equilibrium accuracy but actually degraded far-from-equilibrium smoothness (FSD increased from 80.2 to 97.4).
>
> There could be two explanation to this: either we still haven't reached the interpolation threshold to see the second descent, or there are some fundamental constraints that disallow MLIP smoothness's double descent. We believe that the latter is more likely to be true because far-from-equilibrium configurations represent an out-of-distribution regime where training data is sparse. In such sparse regimes, one cannot rely on implicit regularization from training dynamics. Therefore, explicit regularizations are needed, and by applying explicit, BSCT-guided smoothness regularizations (temperature and smearing width), we successfully scaled our model to 60M parameters, achieving competitive accuracy while maintaining a highly smooth PES (FSD improved dramatically to 43.2).
>
> **Demonstrations of Real Artifacts in Current MLIPs**. We completely agree that visualizing real artifacts is highly useful. We would like to clarify that Figure 5 in our manuscript actually does demonstrate a real artifact found in an MLIP, rather than a synthetic one.
> While Figure 2 uses hypothetical curves (PES1 and PES2) to explain the metric's math, Figure 5 maps the actual force predictions of a real $C_{11}H_{12}NO_2$ molecule from the BSCT-SPICE dataset. It shows a spurious, unphysical spike in the force ratios near a stretched bond length of 2.0 Å, which we trace directly back to abrupt changes in the attention heads. In the revised manuscript, we will update the text and captions to ensure the distinction between the synthetic pedagogical example (Figure 2) and the real model artifact (Figure 5) is unmistakable.
>
> [1] Qu, Eric, et al. "A recipe for scalable attention-based MLIPs: unlocking long-range accuracy with all-to-all node attention." arXiv preprint arXiv:2603.06567 (2026).

---

> > ### Author Rebuttal · Reviewer_2a2d · 2026-04-01
> >
> > I am satisfied with the answers and keep my score.

---

### Official Review · Reviewer_FdVe · 2026-03-13

**Soundness:** 2
**Presentation:** 3
**Significance:** 3
**Originality:** 3
**Overall Recommendation:** 4
**Confidence:** 4

**Summary:**

This paper proposes the bond smoothness characterization test (BSCT), a novel evaluation method for machine learning interatomic potentials (MLIPs). The key motivation is that even when MLIPs achieve low prediction errors in energy and forces, the underlying potential energy surface (PES) may still be physically inaccurate, exhibiting non-smooth or erratic behavior. To assess this, BSCT reduces the PES to a one-dimensional slice by selecting a specific bond, partitioning the structure along that bond, and displacing the two fragments relative to each other to trace out a 1D cross-section of the PES. A novel metric, force smoothness deviation (FSD), is introduced to quantify the smoothness of the resulting profile. Building on the insights gained from BSCT, the authors further introduce a novel model architecture. The effectiveness of both the proposed evaluation method and the model is demonstrated through experiments covering multiple aspects.

**Compliance With Llm Reviewing Policy:**

Affirmed.

**Final Justification:**

I thank the authors for their detailed responses.

All of my concerns have been adequately addressed, and I am satisfied with the responses provided. I have also checked the discussions with the other reviewers and found no major issues. **I therefore maintain my score.**

**Key Questions For Authors:**

1. Phonon calculations inherently require an accurate PES, as phonon frequencies are derived from second derivatives of the energy. As mentioned in the main manuscript, benchmarks such as the MDR phonon benchmark already evaluate this aspect. Could the authors clarify what types of PES artifacts BSCT can detect that phonon-based evaluations would miss, and in what scenarios BSCT would be more informative?
2. The BSCT evaluation requires density functional theory (DFT) calculations as reference. If the model being evaluated was trained on data generated with different DFT calculation settings (*e.g.*, different functionals or basis sets), this mismatch could introduce systematic discrepancies in the FSD results that are unrelated to the model's actual PES quality. Have the authors investigated the sensitivity of BSCT results to such DFT setting mismatches?
3. The FSD metric itself appears reasonable, but its application through 1D bond scanning may be unnecessarily restrictive. A multi-directional perturbation scheme, analogous to the atomic displacements used in phonon calculations, could provide a more comprehensive assessment of PES smoothness. Have the authors considered such extensions?
4. The differentiable k-NN mechanism seems broadly useful and potentially applicable to other MLIP architectures beyond the one proposed here. Have the authors considered applying it to existing models to isolate its effect? More generally, separating the evaluation contribution (BSCT and FSD) from the architectural contribution (diff-kNN) more clearly could strengthen both lines of work.
5. What do the authors consider to be the main limitations of the current work, and what are their plans for addressing these?

**Limitations:**

The authors do not discuss the limitations in the paper. The following points could further strengthen the work:
* Could the authors discuss the applicability of BSCT to periodic or solid-state systems? Given that MLIPs are widely used for crystalline and amorphous materials, addressing this would broaden the impact of the proposed evaluation method.
* The relationship between BSCT and existing PES evaluation approaches, particularly phonon-based benchmarks, should be discussed more explicitly to help position BSCT within the broader evaluation landscape.
* Minor presentation issues, such as redundant acronym definitions, should be cleaned up in the revised manuscript.

The paper presents a well-motivated evaluation method and a complementary model architecture supported by solid experiments. While concerns remain regarding the physical realism and generality of BSCT, the core contribution is valuable. Considering the strengths and weaknesses discussed above, ***I am leaning towards a weak accept for this paper.***

**Strengths And Weaknesses:**

Strengths:
* The motivation behind the proposed evaluation method is compelling. I agree with the authors' observation that low prediction errors in energy and forces do not necessarily guarantee a physically accurate PES, and this is a problem worth addressing.
* The authors go beyond proposing an evaluation metric by introducing a new model architecture informed by their findings. The experimental evaluation of the proposed model is thorough, including ablation studies that clearly demonstrate the contribution of individual components. Specifically, the proposed differentiable k-NN mechanism appears to be a broadly useful contribution to the MLIPs community, potentially applicable beyond the specific architecture presented in this paper.

Weaknesses:
* The physical realism of the BSCT scenario is questionable. While non-equilibrium configurations do arise during molecular dynamics simulations, BSCT constructs geometries by rigidly displacing one fragment while keeping the rest frozen, producing configurations that may not correspond to typical physical pathways. It is unclear whether model behavior on such artificial configurations translates to practical reliability.
* The applicability of BSCT to solid-state systems is unclear. Since the method relies on bond-based partitioning, it appears primarily suited for molecular systems, yet MLIPs are widely used for crystalline and amorphous materials as well.

---

> ### Author Rebuttal · Authors · 2026-03-30
>
> We thank the reviewer for their feedback. Below, we address the reviewer’s questions and weaknesses.
>
> **Physical Realism & Multi-Directional Perturbations.** We agree that rigidly displacing a fragment does not emulate a realistic reaction pathway and that multi-directional displacements represent a broader configurational space. However, this design is an intentional choice for isolating MLIP artifacts.
>
> BSCT is designed as a diagnostic tool rather than a simulation of physical reaction pathways. A 1-D path corresponding to a bond dissociation has a strong theoretical guarantee from quantum chemistry: the potential energy curve should be smooth and contain slow-varying derivatives. If we were to use a more realistic reaction pathway (e.g., NEB-optimized pathways), the ground-truth PES would contain rapidly varying derivatives, making it difficult to distinguish between true physical variations and erroneous MLIP artifacts. We chose 1-D bond dissociation because it offers a clean, mathematically smooth testbed to evaluate the MLIP. While multi-directional perturbations provide a broader assessment, they lack this strict guarantee of monotonic/smooth behavior far from equilibrium and induce significant computational overhead.
>
> We emphasize that although BSCT's 1-D nature makes it a necessary rather than sufficient condition for high-dimensional PES smoothness, it is first-of-its-kind to provide an explicit, computationally efficient test of far-from-equilibrium smoothness that serves as an early indicator of MD instability.
>
> **BSCT vs. Phonon-Based Evaluations.** Phonon evaluations (like MDR or Matbench $\kappa_{SRME}$) probe near-equilibrium smoothness using the harmonic approximation and small atomic displacements (typically 0.01–0.03 Å). In contrast, BSCT stretches and compresses bonds by several Angstroms to probe far-from-equilibrium smoothness.
>
> An MLIP may overfit the near-equilibrium harmonic well perfectly (yielding excellent phonon scores) while exhibiting artificial discontinuities, extrema, or inflection points on far-from-equilibrium conformations. BSCT detects these high-energy, out-of-distribution artifacts that phonon tests fundamentally miss.
>
> **Applicability to Crystalline/Amorphous Systems (Limitations).** The reviewer correctly notes that defining a "bond" is non-trivial in crystalline or amorphous materials, which limits BSCT’s direct applicability to these systems. However, while BSCT is currently constructed for molecular systems, the architectural insights derived from optimizing FSD transfer readily transfer to solid-state domains. For instance, applying our smoothness-oriented designs to MinDScAIP on the MPTrj dataset achieved competitive stability predictions and phonon accuracy on Matbench Discovery (Table 6). Extending the core principles of BSCT to solid-state systems remains an important direction for future work.
>
> **Sensitivity to DFT Settings.** We have not systematically tested BSCT’s sensitivity to different DFT settings and will acknowledge that in the revised manuscript. Nonetheless, the FSD metric is designed to be robust to slight changes in reference value. In Equation 4, FSD is defined using the logarithmic derivative of the force norm ratio. By evaluating the relative rate of change (curve shape) rather than absolute magnitude, FSD is insensitive to constant scaling factors and resilient to baseline shifts caused by different DFT functionals or basis sets. Finally, we highlight that standard evaluations (e.g., phonon calculations, regression errors) similarly depend on underlying DFT references.
>
> **Applying Diff-kNN to Existing Models.** Integrating Diff-kNN into existing foundational models is a promising direction that clearly separates our architectural contribution from our evaluation contribution. Because models like MACE are engineered specifically around radius-graph message passing, swapping their graph construction to Diff-kNN requires non-trivial architectural changes and retuning. Nevertheless, Diff-kNN has already shown practical utility as a modular component in concurrent research; for example, Qu et al. [1] recently utilized it to achieve state-of-the-art results on the OMol25 benchmark. This underscores Diff-kNN as an independent, modular mechanism for the broader MLIP community.
>
> **Main Limitation and Next Step.** We consider the main limitation of our method to be its current reliance on a clean molecular graph, which may be absent for more complex systems. While crystalline and amorphous systems are reserved for future work, we plan to extend the BSCT framework to organometallic complexes and clusters (e.g., OMol25 dataset), where we must go beyond naive molecular graphs.
>
> **Minor issue.** We will fix the presentation, including redundant acronym definitions, in the revision.
>
> [1] Qu, Eric, et al. "A recipe for scalable attention-based MLIPs: unlocking long-range accuracy with all-to-all node attention." arXiv preprint arXiv:2603.06567 (2026).

---

> > ### Author Rebuttal · Reviewer_FdVe · 2026-04-03
> >
> > Thank the authors for the responses.
> >
> > The authors have adequately addressed my concerns. I will finalize my recommendation after reviewing the other discussions.

---

### Decision · Program_Chairs · 2026-04-30

**Decision:**

Accept (regular)

**Comment:**

This paper has received four good quality reviews with similar assessments of the paper: all of them highlight that the paper is technically sound, well written and it offers interesting contributions. The reviews also largely share similar concerns, in particular the observation that the proposed diagnostic tool is limited in the sense that it "only" provides insight on a necessary element of the evaluation of the potential energy surface. This aspect has been discussed at length in the discussion and the reviewers appear satisfied with the authors' responses. My own assessment is that even if a contribution is limited, as long as it is technically sound and useful to some extent, the contribution is valuable. In this particular, the authors justification is that the limitation is compensated by the mathematical and physical robustness of the diagnostic tool.

All things considered, my recommendation is to accept this paper for publication, as this is a technically sound paper, well-written and interesting to at least a portion of the ICML community interested in atomic systems.

I encourage the authors to incorporate the reviewers feedback and their own insights from the rebuttal. Especially, I think it would be important to include a discussion of the limitations of the present work, which is currently missing as highlighted by some reviewers. There are certain limitations that have been discussed during the rebuttal and, in my view, their inclusion in the paper would only strengthen it. Finally, as a minor piece of feedback from me, I encourage the authors to review the bibliography list to ensure that it renders as desired, for example regarding capitalisation (we currently find "dft" or "oc20").